## METHOD

# PAUSE: principled feature attribution for unsupervised gene expression analysis

Joseph D. Janizek[1,2], Anna Spiro[1], Safiye Celik[3], Ben W. Blue[4], John C. Russell[4], Ting-I Lee[4], Matt Kaeberlin[4,5] and Su-In Lee[1*]

*Correspondence:
suinlee@cs.washington.edu

[1] Paul G. Allen School of Computer Science and Engineering, University of Washington, Seattle, USA
[2] Medical Scientist Training Program, University of Washington, Seattle, USA
[3] Recursion Pharmaceuticals, Salt Lake City, USA
[4] Department of Pathology, University of Washington, Seattle, USA
[5] Department of Genome Sciences, University of Washington, Seattle, USA

**Abstract**

As interest in using unsupervised deep learning models to analyze gene expression data has grown, an increasing number of methods have been developed to make these models more interpretable. These methods can be separated into two groups: post hoc analyses of black box models through feature attribution methods and approaches to build inherently interpretable models through biologically-constrained architectures. We argue that these approaches are not mutually exclusive, but can in fact be usefully combined. We propose PAUSE (https://github.com/suinleelab/PAUSE), an unsupervised pathway attribution method that identifies major sources of transcriptomic variation when combined with biologically-constrained neural network models.

**Keywords:** Transcriptomics, Gene expression, Deep learning, Explainable AI, Unsupervised learning, Feature attribution

## Background

As technologies to measure gene expression in biological samples have advanced over the last several decades, the tools and methods to extract biological meaning from these high-dimensional measurements have developed in parallel [1]. The increasing number of available samples in large transcriptomic compendia has enabled deep learning as a viable option for reference mapping [2], data integration [3], dimensionality reduction and visualization [4]. In particular, *unsupervised* machine learning approaches to solve these problems have become popular. For instance, tools like scVI use variational autoencoders (VAEs) to help analyze single cell RNA-seq data [5]. Similarly, variational autoencoders have been used to learn low-dimensional embeddings of bulk cancer RNA-seq data, improving the prediction of drug response [6]. Deterministic autoencoders have also been used in transcriptomic analyses, including the application of denoising autoencoders to large *Pseudomonas aeruginosa* and yeast datasets [7, 8]. While these deep learning approaches all are capable of modeling complex systems with high fidelity, they unfortunately share the

drawback of lacking interpretability. Their latent embeddings do not have inherent biological meaning, and further methods are required to understand mechanisms captured by these models [9].

Two competing trends have emerged to "open the black box" and elucidate the biological mechanisms learned by these models, and have so far been framed as orthogonal approaches. The first trend could be described as post hoc interpretability, and aims to identify which genes were the most important contributors to each latent variable learned by an autoencoder. A variety of methods have been used to do this interpretation, and range from the use of feature attribution methods [10–12] to the direct analysis of weights in shallower autoencoders. For instance, Dincer et al. [13] apply the feature attribution method Integrated Gradients [12] to the latent variables of VAEs to identify the most important genes for each latent dimension, while Way et al. [14] measure the magnitude of each input weight to shallow, non-linear autoencoders. Likewise, in single cell analysis, Svensson et al. [15] modify a popular VAE architecture to have a linear decoder, allowing enrichment tests to be run using the magnitude of weights in that linear decoder. Despite the successes of these approaches, these types of analyses may be very difficult as autoencoders are known to learn entangled representations, meaning that each latent variable may capture multiple biological processes [16]. Furthermore, generating feature attributions for dozens or hundreds of latent variables may be computationally inefficient.

Another recent trend could be described as *biologically-constrained modeling*, and aims to create models with latent spaces that are inherently interpretable. These models use prior information to define sparse connections between input nodes corresponding to genes, and latent nodes corresponding to biological pathways (or other pre-defined groups). These biologically-constrained networks have been used in a supervised setting to improve the prediction of survival or treatment resistance from cancer gene expression or mutational status [17, 18], and to improve Genome Wide Association studies by aggregating the effects of single nucleotide polymorphisms (SNPs) into SNP sets [19]. In particular, a variety of recent works have proposed using biologicaly-constrained autoencoders to model gene expression data [20–22].

Our paper aims to demonstrate that these two trends in biological interpretability are not mutually exclusive, and that principled attribution methods can improve the analysis of unsupervised models of gene expression analysis by quantifying the importance of pathways. We first outline a general workflow for unsupervised analysis of gene expression data, comparing classical linear approaches like principal component analysis (PCA) to more contemporary deep learning-based approaches. This outline allows us to identify a step in the workflow that has been neglected by previous methods. We then propose a novel, fully-unsupervised attribution method and demonstrate how it can be used to identify important pathways in transcriptomic datasets when combined with biologically-constrained autoencoders. This allows for *fully unsupervised* analysis of gene expression data. We next show how existing, feature-level attribution approaches still provide useful information for annotated, unsupervised models. Finally, we apply our approach to a large transcriptomic compendium of post-mortem brain data from patients with Alzheimer's disease and demonstrate how our approach can identify Mitochondrial Respiratory Complex I as a potential drug target for this disease.

## Results

### Overview of PAUSE approach

To understand how game-theoretic attributions can improve the unsupervised analysis of gene expression data, it is first helpful to understand a representative unsupervised workflow (Fig. 1a) [23, 24]. In the past, researchers have used linear approaches like Principal Components Analysis (PCA) to (1) learn a low-dimensional representation of gene expression data, (2) rank the latent dimensions according to the amount of variance in the original data explained by each dimension, and finally (3) interpret the biological meaning of the most important dimensions. Finding the importance of each latent dimension is straightforward in PCA, as the algorithm inherently arranges the coordinates in descending order of the amount of variance in the data explained by each component [25]. Interpreting the biological meaning of the coordinates is fairly straightforward as well, as the magnitude of the gene loadings for each component identify the important genes, which can then be tested for enrichments in particular biological processes using tools like Enrichr [26] or StringDB [27].

While deep learning-based autoencoders are able to reconstruct gene expression with high fidelity, they fall short at steps (2) and (3) in the workflow described above. Both the relative importance of the different latent dimensions, and the biological meaning of these dimensions are opaque in deep autoencoders. "Interpretable" autoencoders (Fig. 1b) aim to improve step (3) for deep learning models by constraining the learned representation so that the latent dimensions correspond to known biological pathways. By "interpretable" autoencoder model, we refer to any model that learns a latent representation with dimensions that correspond to biological pathways or functions (Fig. 1b). These models can learn pathway embeddings that are complex, non-linear functions of the input genes, but are restricted in that each learned latent pathway dimension only incorporates information from genes that are pre-annotated to that pathway using databases like Reactome [28]. This restriction is encoded either as a hard constraint using sparse masks across layers [22], or as a soft constraint using regularization [20] (see the "Model architectures" section for more details).

While interpretable autoencoder models have latent nodes corresponding to biological pathways, there is no clear-cut way to identify which pathways are the most important in a dataset. Our approach, *p*rincipled *a*ttribution for *uns*upervised gene *e*xpression analysis (PAUSE), aims to improve the utility of "interpretable" deep autoencoder models using techniques from the area of feature attribution (Fig. 1c). While the eigenvalues in PCA correspond to the amount of variance in the original expression space explained by that component, deep learning-based autoencoders lack an obvious correspondence revealing how much variance is explained by each latent dimension. Using approaches from game theory for credit allocation among players in cooperative games, we derive a novel pathway attribution that can be thought of analogously to the eigenvalues in PCA, in the sense that this attribution value shows how much variance in the original gene expression space is explained by each latent pathway. By posing the reduction in reconstruction error as the reward to be allocated in a cooperative game in which the pathways are the players (see the "Methods" section for more details), solution concepts like the Shapley value [29] or Aumann-Shapley value [30] can be used to provide *pathway attributions*. While we primarily applied our pathway attributions to

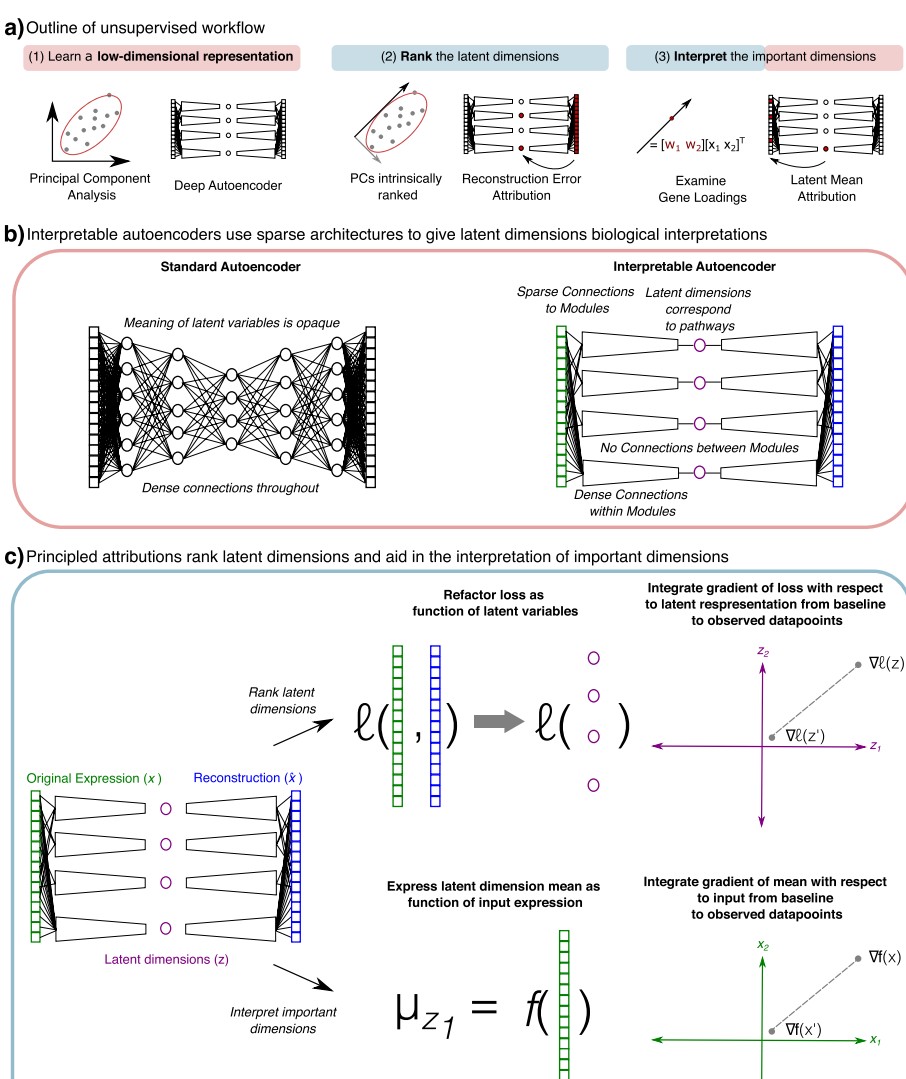

**Fig. 1** Principled attributions complement biologically-structured networks to create more interpretable unsupervised models. **a** An outline of a general workflow of unsupervised analysis of gene expression data, comparing classical linear approaches (left in each subpanel) and deep learning approaches (right in each subpanel). (1) First, a dimensionality reduction algorithm such as PCA (left) or a deep autoencoder (right) is applied to a dataset of gene expression values to learn a low-dimensional representation. (2) After this low-dimensional representation is learned, the learned dimensions must be ranked by their importance. This ranking is inherently provided in PCA, which sequentially maximizes directions of unexplained variance in the data. There currently are no principled approaches to provide this ranking in deep models, which is the gap in the literature filled by our novel loss attribution. (3) After finding the most important latent dimensions, the biological meaning of these dimensions is interpreted. In PCA (left), the contribution of different genes to each dimension can be found by examining the magnitude of the gene loadings. For deep learning models, feature attribution methods can be applied to determine gene contributions. **b** In standard autoencoders, the learned latent variables have opaque meanings, as their relationships with input genes are unknown. Biologically-constrained models increase the interpretability of latent variables by using sparse connections or regularization to ensure that latent dimensions correspond to pre-defined pathways. **c** We apply principled attribution methods to help rank the latent dimensions of autoencoder models and to interpret the biological meaning of the most important dimensions. Attributing the model's reconstruction error to the latent dimensions quantifies the importance of each latent dimension. Attributing the output of each latent dimension to the input genes quantifies the contribution of each input gene to each learned pathway

biologically-constrained models, they can be applied to any unsupervised autoencoder to identify the most important latent dimensions (see Additional file 1: Fig. S1, Methods)

Additionally, although the latent factors of biologically-constrained autoencoders are more interpretable than the latent factors of standard autoencoders, in the sense that they only represent the expression of genes within their annotated pathway, these latent variables are often not fully interpretable on their own. For example, the pathway module variational autoencoder (pmVAE) architecture [22] learns multiple latent variables in the bottleneck layer of each pathway module, which helps to encode more biologically relevant information. However, this approach provides no way for understanding the differences between the different latent nodes within a given pathway module. Are these latent variables sub-pathways supported by different genes with different expression patterns? Furthermore, if there are important pathways in an expression dataset that are not represented in the knowledge base used to define the model architecture, can we still identify these pathways? We propose generating *gene attributions*, again using feature attribution techniques based on the Aumann-Shapley value, to identify the genes contributing to important learned pathway embeddings, or to identify the genes contributing to densely connected auxiliary pathways (Fig. 1c).

### Pathway attributions accurately identify major sources of variation

To empirically validate that the pathways identified by our pathway attributions correspond to the most important sources of variation in real gene expression datasets, we adapt two benchmarks from the feature attribution literature [31]. The first is an *imputation* benchmark, and measures the extent to which the reconstruction error of a trained biologically-constrained autoencoder model increases when the learned embeddings for each pathway are replaced with an uninformative mean imputation. The better an attribution method ranks important pathways, the more quickly the reconstruction error will increase. The second benchmark is a *retrain* benchmark, and measures the extent to which pathways identified as important can reconstruct gene expression space when used to train a new model. Attributions that do a better job of ranking pathways will decrease the reconstruction error more quickly (Fig. 2).

Previous approaches to identifying important pathways in autoencoder models have been limited in a variety of ways. For example, previous papers have used supervised metrics like logistic regression accuracy or Bayes factors to identify important pathways [20, 22]. While these approaches have been used to successfully identify important pathways, calculating these metrics depends on having labeled data, which means that this analysis is not truly unsupervised. In terms of fully unsupervised approaches, Higgins et al. [16] have proposed finding important latent factors of variation by measuring the Kullback-Leibler (KL) divergence between the learned latent distribution in VAEs and the prior distributions. While this approach does not require labeled data, it can only be applied to VAE models, and can not be applied to standard, deterministic autoencoders. Another fully unsupervised approach to ranking important pathways examines the L2 norm of the weights connecting each latent pathway to the reconstruction output; however, this metric is obviously limited to models with linear decoders [21]. Unlike all of these approaches, our proposed pathway attribution (see the "Methods" section) is both

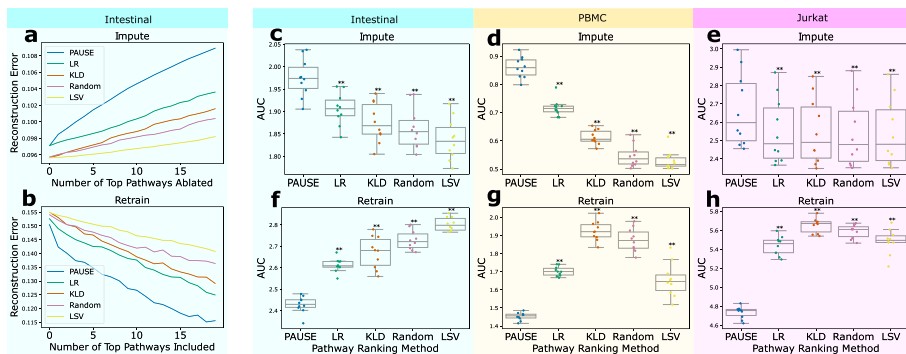

**Fig. 2** PAUSE pathway attributions accurately identify the major sources of variation in single cell datasets. To verify that our novel pathway attribution method ("PAUSE" in the plots above) is capable of identifying the major sources of transcriptomic variation across a variety of datasets, we apply two benchmarks of pathway identification. The first (**a**), termed our *impute* benchmark, measures how much the reconstruction error of a biologically-constrained autoencoder model increases as important pathways are replaced with an uninformative, imputed baseline. Better methods will increase the error faster, leading to a larger area under the curve (AUC). The second benchmark (**b**), termed our *retrain* benchmark, measures how well a model can reconstruct the observed expression when retrained using only the most important pathways. Better methods will decrease the error faster, leading to a smaller AUC. The AUCs for both the impute (**c**–**e**) and retrain (**f**–**h**) benchmarks are shown for ten separate train/test splits across three separate single cell gene expression datasets (intestinal cells, peripheral blood monocytes, and Jurkat cells). In each experiment, the PAUSE loss attribution method significantly outperforms other methods. The other methods shown here are logistic regression score (LR), Kullback-Leibler divergence (KLD), random ranking, and latent space variance (LSV). The boxes in **c**–**h** mark the quartiles (25th, 50th, and 75th percentiles) of the distribution, while the whiskers extend to show the minimum and maximum of the distribution (excluding outliers)

fully unsupervised, meaning it requires no labeled data, and model agnostic, meaning it can be applied to any model regardless of architecture or implementation details.

The three datasets used in the two benchmarks were selected because they have labels corresponding to known perturbations applied to the cells, which allowed us to compare the information gleaned from our fully unsupervised analysis to that highlighted by supervised methods. These datasets have also been extensively analyzed in previous, related work, and include a dataset of peripheral blood monocytes (PBMC) stimulated with interferon-$\beta$ [32], a dataset of intestinal epithelial cells stimulated with *Salmonella enterica* and *Heligmosomoides polygyrus* [33], and a dataset of Jurkat cells stimulated with a combination of anti-CD3/anti-CD28 antibodies to induce T cell activation [23] (see the "Methods" section for more details, including data preprocessing). The particular model architecture trained for this benchmark was a pathway module VAE (pmVAE), which is a sparse variational autoencoder model with deep, non-linear encoders and decoders [22]. Unlike standard autoencoders, which have dense connections between every gene and each latent node in the first encoder layer, and dense connections between every latent node in each successive layer, the pmVAE uses sparse connections to define "modules" of nodes that have dense connections between nodes within the modules and no connections between nodes in different modules. The first layer of the encoder (and the last layer of the decoder) is a sparse layer that connects each gene only to the modules corresponding to the pathways to which it has been annotated. In this model, each module can be thought of as a separate dense autoencoder, which all sum together at the output layer. Across both benchmarks and all three datasets, our pathway attribution method consistently identified the most important sources of dataset-wide

transcriptomic variation better than previous approaches (two-sided Wilcoxon signed rank test, Bonferroni corrected $p = 7.81 \times 10^{-3}$, statistic $= 0.0$, see Fig. 2). When we repeat this benchmark with three additional single cell RNA-seq datasets, we find the same results (see Additional file 1: Fig. S2).

In addition to validating the performance of PAUSE on the pmVAE model, we also evaluated the performance of PAUSE on other architectures using this benchmark. Remarkably, we found that our pathway loss attributions outperformed directly examining the L2 norm of the decoder weights in an architecture with a linear decoder (see Additional file 1: Fig. S3). Next, we looked at the results of the imputation benchmark on an unconstrained VAE that does not incorporate *any* pathway information. Again, we see that PAUSE still significantly outperforms other methods at identifying pathways whose ablation leads to an increase in reconstruction error (see Additional file 1: Fig. S4). Interestingly, we observe that in this case, when the VAE does not have biological latent space constraints and is optimized only to minimize divergence from an isotropic Gaussian prior and reconstruct the input, KL divergence does quite well at identifying important pathways. However, PAUSE still significantly outperforms KL divergence, even in the absence of a pathway-based model architecture. In conclusion, our benchmark demonstrates that our pathway attribution method is able to accurately identify the major sources of transcriptomic variation across different datasets and autoencoder architectures, showing the benefit of using prinicpled attributions in conjunction with biologically-constrained models.

### Pathway attributions identify biologically relevant pathways

After demonstrating in the previous section that our pathway attributions were capable of accurately identifying the major sources of transcriptomic variation in three scRNA-seq datasets, we wanted to demonstrate that these major sources of transcriptomic variation corresponded to biologically interesting pathways. We therefore compared the top pathways found by our unsupervised approach to the top pathways according to a more conventional, supervised analysis (Fig. 3).

We first considered a biologically-constrained VAE (pmVAE, see the "Methods" section) trained on a dataset of peripheral blood mononuclear cells (PBMCs), where the PBMCs were either untreated as controls or stimulated with interferon-$\beta$ [32]. The latent nodes of this model were defined using Reactome pathways [28]. Because this is a well-studied dataset with a known ground truth perturbation, we know that pathways related to interferon-$\beta$ (IFN-$\beta$) stimulation should be important, and because we have control/stimulated labels for the cells, we can compare the pathways identified as important by our unsupervised analysis with the pathways identified by supervised analysis. For a supervised metric of pathway importance, we considered the accuracy attained by a logistic regression model trained on the pathway latent nodes to differentiate stimulated and controlled cells [22].

We find that our fully unsupervised pathway attributions are able to identify many of the same pathways as supervised approaches, without requiring access to labeled data (see Fig. 3a). For example, we see that the Reactome pathways "Cytokine Signaling in Immune system," "IFN Signaling," and "IFN alpha/beta signaling," are in the list of top ten pathways identified by both approaches. More interestingly, we can also

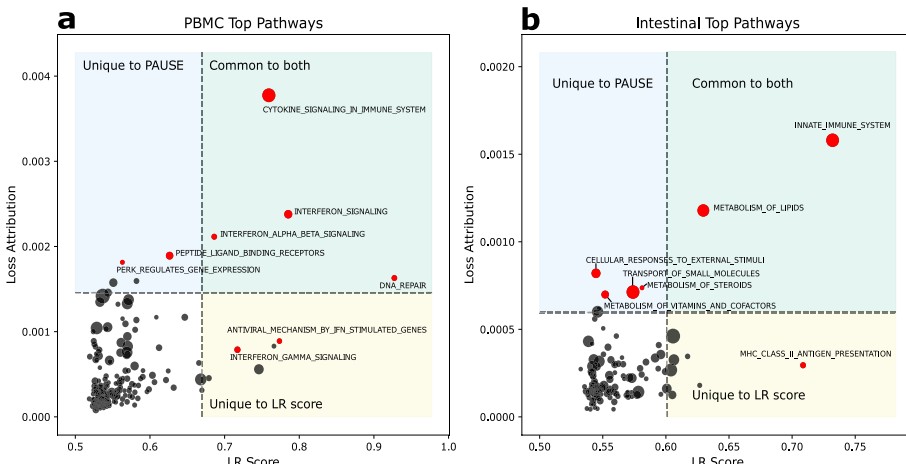

**Fig. 3** PAUSE loss attribution reveals biologically relevant pathways that overlap with pathways found using a supervised approach. Comparison of top pathways found using PAUSE loss attribution (y-axis, unsupervised) and logistic regression (LR) score (*x*-axis, supervised) in PBMC (**a**) and Intestinal (**b**) datasets. Gene sets in the blue rectangles are those ranked in the top 10 only by PAUSE, those in the yellow rectangles are ranked in the top 10 only by LR score, and those in the green rectangles are ranked in the top 10 by both methods. The size of the dots indicates gene set size

examine where the pathway rankings are discordant. For example, "IFN Gamma Signaling" is a top pathway according to the supervised approach, but not according to our unsupervised rankings. This demonstrates why our unsupervised approach can complement the supervised approach even when labels are present. Although certain genes are regulated by both type I and type II IFNs, other genes are expected to be selectively regulated by either type I or type II IFNs [34]. Looking at our unsupervised attributions shows that while there may be significant enough differences in expression of genes in the "IFN Gamma Signaling" pathway to differentiate stimulated and unstimulated cells, this pathway is not one of the major sources of variation in the dataset, at least when compared to pathways more directly related to IFN-$\beta$ signaling. Therefore, in addition to being useful when labeled data is not present, our unsupervised attributions can provide additional utility even when labels are present by explicitly quantifying the amount of variance in the observed data explained by each pathway.

We next considered a VAE (again with latent modules defined by Reactome pathways) trained on a dataset of mouse intestinal epithelial cells where control cells were untreated, and stimulated cells had been exposed to the parasitic roundworm *Heligmosomoides polygyrus* [33]. Again, there is substantial overlap in the top pathways identified by our unsupervised approach and supervised metrics (Fig. 3b). For example, the Reactome pathway "Innate Immune Response" is the most important pathway according to both supervised and unsupervised attributions. Next, we again can look to see where these approaches are discordant in their rankings. Examining the top ten pathways according to supervised attributions, we see that the pathway "MHC Class II Antigen Presentation" is a top pathway, despite not being highly ranked by our unsupervised metric (Fig. 3b). This pathway is particularly biologically plausible as an important process in this dataset, as the MHC Class II complex is constitutively

expressed in the upper villi of the small intestine, indicating a potential role as non-professional antigen presenting cells for intestinal epithelial cells [35]. Previous work has shown that in both epithelial cells and in other cell types, expression of the MHC Class II complex is regulated in response to pathogen exposure [36, 37]. The fact that this pathway is highlighted by supervised attributions, but not by our unsupervised approach, shows that these two approaches (supervised and unsupervised) should be considered complementary. While our PAUSE attributions can help identify the pathways responsible for explaining the largest quantities of variation in a given dataset, supervised attributions can help identify specific differences in expression between known groups. Analysis of additional single cell RNA-seq datasets can be found in Additional file 1: Figs. S5–6.

### Gene attributions increase latent node interpretability

In addition to identifying important pathways, the interpretability of biologically-constrained models can be enhanced through the application of gene attribution values. These gene attribution values help identify which gene expression values are important determinants of each learned pathway representation. For example, previous work has analyzed the expression of Jurkat cells, an immortalized human T lymphocyte cell line, when stimulated with anti-CD3 and anti-CD28 antibodies to induce T cell activation [23]. This prior analysis trained a pathway module VAE (pmVAE) to reconstruct the expression of these cells, and found that having multiple nodes in the bottleneck layer of each pathway module led to embeddings that were more discriminative for T cell activation [22]. This raises the question of whether multiple nodes may be necessary in order to distinguish different processes of the biology of T cell activation.

We therefore generated *gene attributions* to understand the expression programs represented by the multiple nodes present within given pathway modules. The two most important genes for each of the four latent nodes in the TCR signaling module were PTPRC and PTPN22 (Additional file 1: Fig. S7). PTPRC enocdes a tyrosine phosphatase commonly known as CD45 antigen, while PTPN22 codes for another protein tyrosine phosphatase commonly known as PEP. These genes are known to play an important, albeit complex, role in proximal TCR signaling, regulating the activity of Src family kinases downstream of T cell receptor activation [38].

Looking at the pairwise correlations between the activations of each of the four latent nodes in the TCR Signaling pathway module over all cells (see Fig. 4a), we see that while many of the pairs of nodes have a high degree of correlation or anti-correlation (potentially indicating redundancy between these nodes), Node 1 and Node 2 have a relatively low correlation (Pearson's $R = 0.11$). We can plot gene attributions for the top two genes for Node 1 (Fig. 4b, c) and Node 2 (Fig. 4d, e) in order to understand what precisely is being represented by the different nodes of the pathway module.

We see that Node 1's output is high when PTPRC's expression is high (Fig. 4b), and low when PTPN22 expression is high (Fig. 4c). This means that in cells where PTPRC and PTPN22 are co-expressed, the contributions of these two genes will cancel out and the latent node's activation will have a low magnitude. Node 2's output, in contrast, is highly negative when PTPRC expression is high (Fig. 4e) and highly negative when PTPN22 expression is high (Fig. 4f). When we visualize all of the cells in the dataset by their

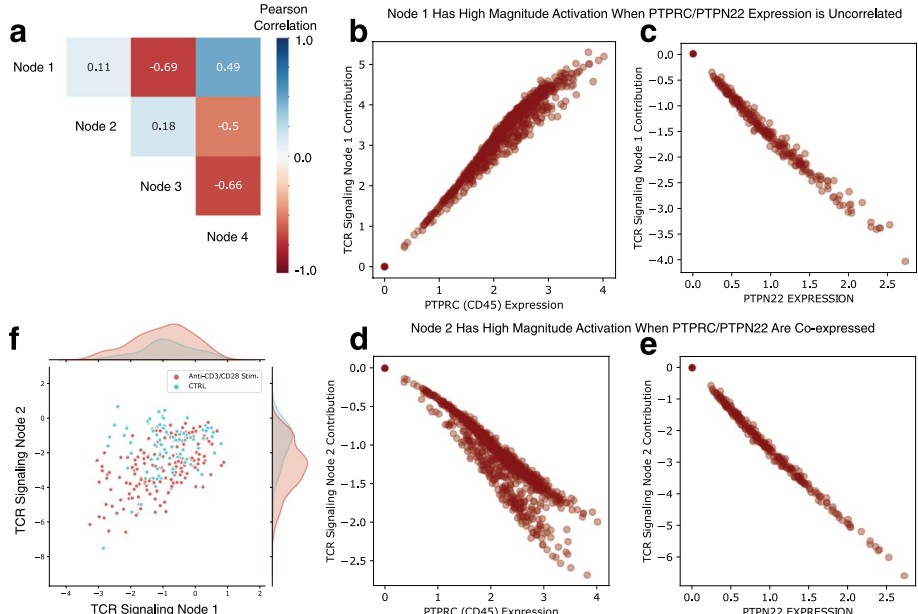

**Fig. 4** Gene attributions increase latent variable interpretability and differentiate TCR signaling pathway expression programs. Gene level attributions help to gain a deeper understanding of the expression programs represented by the latent variables of a biologically-constrained autoencoder trained on a dataset of Jurkat cells stimulated with anti-CD3 and anti-CD28 antibodies. **a** Pairwise correlations between the four latent variables in the pathway module corresponding to T cell receptor signaling. **b–c** Gene attribution dependence plots for the two most important genes, ranked by average magnitude gene attribution over all samples in the dataset, for TCR Signaling Latent Node 1. **d–e** Gene attribution dependence plots for the two most important genes, ranked by average magnitude gene attribution over all samples in the dataset, for TCR Signaling Latent Node 2. **f** Jurkat cells plotted by their embedding in the first two nodes of the TCR Signaling Pathway module. The first node, in which PTPRC and PTPN22 expression are not co-regulated, does not separate cells that have been stimulated by anti-CD3/anti-CD28 antibodies (Wilcoxon rank-sums test statistic $= -0.95$, $p$ value $= 0.342$), while the second node, in which PTPRC and PTPN22 expression levels are highly correlated, does separate cells that have been simulated by anti-CD3/anti-CD28 antibodies (Wilcoxon rank-sums test statistic $= -6.27$, $p$ value $= 3.39 \times 10^{-10}$)

embedding values for these two latent nodes after filtering out cells where the values of these top genes are dropped out (see Fig. 4f), we see that Node 2 separates anti-CD3 and anti-CD28 antibody-stimulated cells (Wilcoxon rank-sums test statistic $= -6.27$, $p$ value $= 3.39 \times 10^{-10}$), while Node 1 does not (Wilcoxon rank-sums test statistic $= -0.95$, $p$ value $= 0.342$). This demonstrates the utility of gene attributions. While previous work found that having multiple nodes in the bottleneck layer of each pathway module led to embeddings that were more discriminative for T cell activation [22], there was no direct way to distinguish which expression patterns the model was using to define each latent node.

In addition to differentiating sub-processes within an annotated pathway module, another use for gene attributions is identifying biological processes learned by unannotated, densely-connected modules. In addition to sparse modules corresponding to annotated biological pathways, densely connected modules can be jointly modeled to capture novel biology. In order to verify that densely connected modules could identify biological expression programs not represented in prior knowledge bases, we modeled a dataset where we know the ground truth perturbations (PBMCs stimulated with IFN-$\beta$). We omitted a group of pathways related to the known perturbation by training a new

model without the corresponding modules for those pathways, but included a module connected to all genes with four nodes in its bottleneck layer. When we used gene attributions to identify the important expression contributors for the latent nodes in the bottleneck layer of this densely-connected, unannotated module, we found that they corresponded to the expected ground truth biology (Additional file 1: Fig. S8). Specifically, the 50 most important genes for each densely connected node were significantly enriched for multiple pathways corresponding to biological ground truth (Additional file 1: Fig. S9).

As a control experiment, we repeated this procedure, but instead of omitting pathways that correspond to known biological ground truth, we omitted pathways that we expect to be unrelated to the given perturbation. As an example, we look at removing the Reactome pathway "Regulation of PLK1 Activity at G2/M Transition," which is involved in regulating a number of cellular proteins during mitosis [28]. We then again used gene attributions to identify the most important expression contributors to the densely-connected nodes (Additional file 1: Fig. S10). In this case, we found no significant pathway enrichments for the top 50 genes for any of the auxiliary nodes. This control experiment shows that densely-connected nodes can specifically represent unknown, novel sources of biological variation, rather than all missing pathway information. Gene attributions allow these densely-connected nodes to be inspected to understand biological signal not represented in prior knowledge. While this experiment does not guarantee that a dense module will always find the most important sources of missing variation, it does demonstrate that it is able to represent important pathways when they are omitted from the prior, and that it will not necessarily represent unimportant pathways that are omitted from the prior.

In conclusion, while pathway attributions help give a high-level view of important processes in gene expression datasets, gene attributions help elucidate particular mechanisms in greater detail. Furthermore, these attributions can help identify novel biology in cases where the prior knowledge used to define the biologically-constrained model is mismatched with the expression programs present in the dataset being modeled.

### PAUSE identifies mitochondrial oxidative phosphorylation as an important process in Alzheimer's brain samples

After testing the PAUSE approach by applying it to a variety of single cell datasets with known perturbations and well-characterized downstream effects, we wanted to use our PAUSE approach to gain insight into the biology of Alzheimer's disease (AD). AD is the most common form of dementia, and accounts for an estimated 60–70% of worldwide cases [39]. Clinically, AD manifests with memory loss and cognitive decline, while neuropathologically AD is associated with amyloid-$\beta$ (A$\beta$) plaques and abnormal tau tangles in the brain [40]. We therefore applied PAUSE to a large collection of postmortem brain RNA-sequencing measurements from patients with AD, assembled by the AMP-AD (Accelerating Medicines Partnership Alzheimer's Disease) consortium. This dataset includes brain samples from a variety of sources, including the Religious Orders Study/ Memory and Aging Project (ROSMAP) [41], Adult Changes in Thought (ACT) [42], Mount Sinai Brain Bank (MSBB), and Harvard Brain Tissue Resource Center (HBTRC) studies (see the "Methods" section for more detail).

Because of the diversity of data sources combined in this dataset, it was essential to account for batch effects so that the embedding learned by our model represented true biological variation rather than technical artifacts (Fig. 5a). When a standard pmVAE model is trained on the data, we see that the latent space separates samples predominantly according to the data source from which the samples were derived (Fig. 5b). To control for these dataset effects, we therefore modified the pmVAE architecture to be a *conditional* pathway module VAE (cpmVAE) (Fig. 5a, Additional file 1: Fig. S11). In addition to the set of gene expression values corresponding to a particular pathway, each module takes as an additional input a vector of labels describing any unwanted sources of technical variation (like data source identity). By explicitly encoding the batch effects in these variables, the model is free to learn biological information in the latent space. When our cpmVAE model is trained on the data, we see that there is less separation in the latent space on the basis of source dataset (Fig. 5c, Additional file 1: Fig. S12). To assess the quality of the embedding learned by our cpmVAE model and further "sanity check" our approach, we showed that our cpmVAE model reconstructed the original

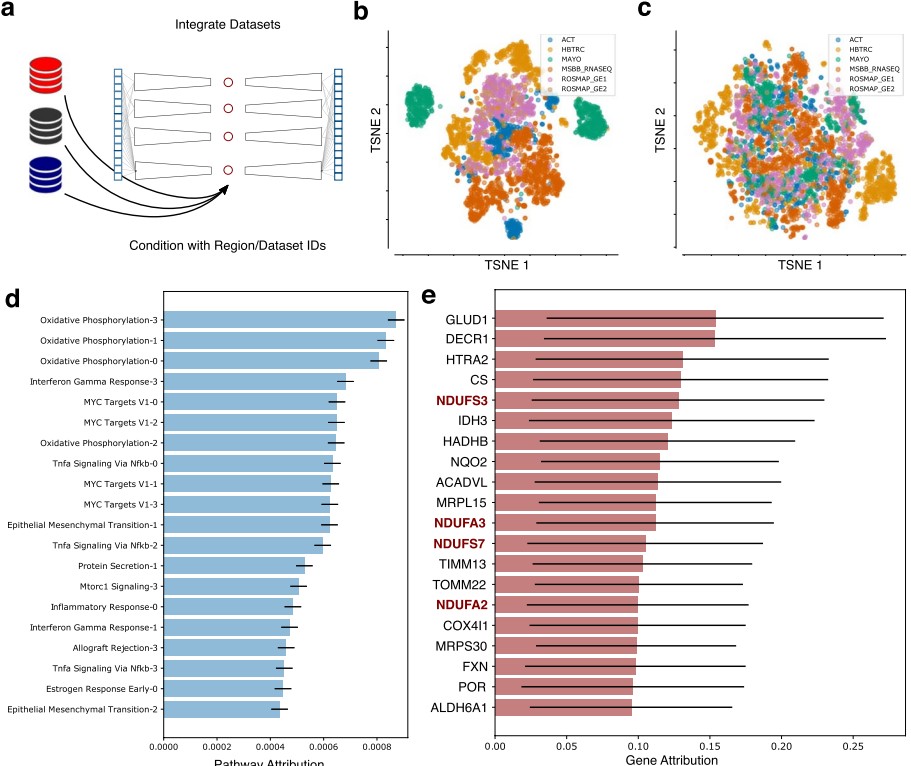

**Fig. 5** PAUSE analysis highlights the importance of Mitochondrial Respiratory Complex I in Alzheimer's disease. **a** A conditional pathway module VAE (cpmVAE) is trained to integrate human post-mortem brain tissue samples from multiple Alzheimer's disease (AD) studies. **b** TSNE plot of the latent space of a pathway module VAE (pmVAE) trained *without* batch effect correction by conditioning on study ID. **c** TSNE plot of the latent space of a cpmVAE trained *with* batch effect correction by conditioning on study ID. **d** Most important pathway latent variables in cpmVAE model according to PAUSE pathway loss attribution. Error bars indicate the standard deviation over local attributions **e** Most important gene expression contributors for top pathway (Oxidative Phosphorylation Node 3). Gene names bolded and in red encode proteins in Mitochondrial Respiratory Complex I. Error bars indicate the standard deviation over local attribution magnitudes

gene expression space more accurately than a standard pmVAE model, had less clustering by dataset source, and was more predictive of the neuropathological phenotype information we had for a small subset of samples than the original gene expression space (Additional file 1: Fig. S12).

When we generate PAUSE pathway attributions to interrogate the latent space learned by our cpmVAE model, we can see that the single pathway explaining the most observed variation in the biological latent space is the Hallmark Oxidative Phosphorylation pathway (Fig. 5d). Oxidative phosphorylation in the mitochondria is one of the major metabolic processes responsible for generating the ATP necessary for neuronal function, and dysregulation and differential expression of this pathway has previously been linked to AD [43]. While primary defects in oxidative phosphorylation are not thought to be likely causes of AD, and mitochondrial dysfunction is in fact thought to be the result of A$\beta$ and tau protein accumulation [44], these changes in expression of oxidative phosphorylation genes likely play an important role in the pathophysiology of AD [45]. Furthermore, the expression of this pathway varies not only between patients with AD and without AD, but also varies greatly across AD patients, suggesting that this pathway may be related to the heterogeneity of clinical phenotypes observed in AD patients [46]. We additionally confirmed that there was significant overlap between the pathways identified by PAUSE and pathways identified by a supervised method on the subset of data for which we had neuropathological phenotype information (see Supplementary Table 1).

After identifying several nodes related to oxidative phosphorylation as the top general processes in this dataset, we used gene attributions to further interrogate the specific function represented in the top pathway node, which is Oxidative Phosphorylation Node 3. We can visualize the individual gene attributions for this node using a summary plot (Fig. 5e). When we look at the top 20 genes by gene attribution, we see four genes, NDUFS3, NDUFA3, NDUFS7, and NDUFA2, that are all part of mitochondrial respiratory Complex I, which is responsible for the oxidation of NADH in the mitochondria [47].

**Experimental validation of PAUSE analysis identifies mitochondrial Complex I as a potential AD therapeutic target**

Unlike the single cell datasets with clear ground truth perturbations and well-studied downstream effects examined in the prior analyses, the biology underlying the differences in gene expression in real-world samples from patients with AD is less clearly characterized. Therefore, in order to verify that the genes and pathways identified by PAUSE are biologically-relevant, we could not reference a known ground truth, and had to experimentally validate our findings.

To gain insight into the biological relevance of the genes identified by our PAUSE analysis of the Alzheimer's brain expression data, we used the nematode *C. elegans*, a well-established animal model of A$\beta$ proteotoxicity [48]. We conducted experiments with GMC101, a transgenic worm line displaying an age-associated aggregation of human A$\beta$ 1−42 peptide resulting in rapid onset of age-associated paralysis. A stringent reciprocal best hits (RBH) approach (BLAST *e*-value $\leq 10^{-30}$; details in the "Methods" section) was used to identify nematode orthologs for human genes. This assay can be used to test the effects of various genes on A$\beta$ proteotoxicity, as transgenic A$\beta$1−42 accumulates over time in the bodywall muscle of GMC101 worms, leading to a paralysis phenotype

mediated by A$\beta$ proteotoxicity [49]. Gene expression perturbations that significantly increase the observed time to paralysis therefore are demonstrated to impact A$\beta$1–42 toxicity in this model organism.

We used RNAi via bacterial feeding to reduce expression of 13 *C. elegans* genes encoding homologs of human Complex I proteins in animals expressing toxic A$\beta$. This resulted in a striking delay in paralysis relative to control animals treated with empty vector (EV) RNAi (Fig. 6) (details in the "Methods" section). Suppression of paralysis by knockdown of Complex I genes was comparable, or often exceeded, the suppression of paralysis by our positive control, RNAi knockdown of the insulin-like receptor gene daf-2 (Fig. 6), one of the strongest known suppressors of A$\beta$ toxicity in worms [50].

We observed that RNAi knockdown conferred protection against amyloid beta toxicity, as evidenced by delayed paralysis. This may indicate that mitochondrial Complex I activity contributes to the pathological consequences of high A$\beta$ burden. Alternatively, knock-down of mitochondrial Complex I components could induce a protective response that attenuates A$\beta$ toxicity. This will be an important topic to address experimentally through future study.

## Discussion

By combining principled attributions with pathway-based representations, our PAUSE approach enables the interpretable and fully unsupervised analysis of gene expression datasets. In both the single cell and the bulk RNA-seq datasets analyzed, we found that the major sources of variation quantified by our pathway attribution approach corresponded to biologically-meaningful processes, as confirmed by known ground-truth perturbations in the single cell analyses and by experimental validation in the AD dataset. Unlike existing methods, our pathway attributions do not depend on having labeled data to identify important pathways. This enables fully unsupervised data exploration, and can complement supervised analyses when labels are present. Furthermore, this

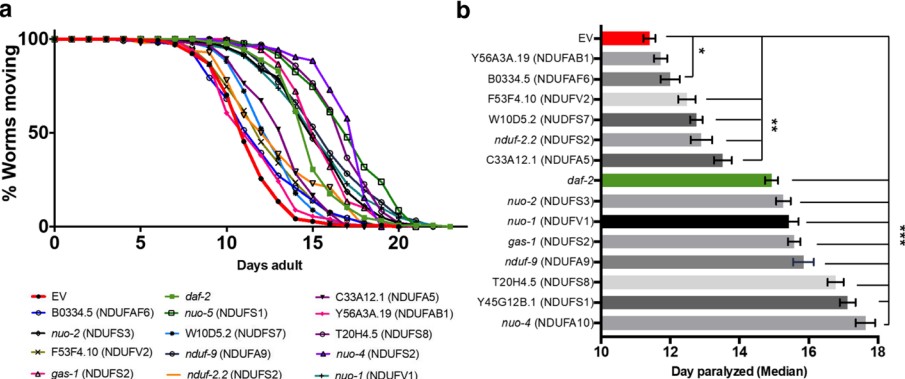

**Fig. 6** *C. elegans* A$\beta$ proteotoxicity assay validates importance of mitochondrial Complex I genes. **a** Paralysis curves for the reciprocal best orthologs of human Complex I genes. All tested RNAi conditions significantly suppress paralysis. **b** The same data as in **a** plotted as the median day of paralysis for each population of worms. Half of the conditions showed even stronger suppression than daf-2 RNAi conditions. Error bars indicate standard error of the mean across experiments. * *p*-value < 0.05. ** *p*-value < 0.01. *** *p*-value < 0.001

approach can be applied to arbitrarily deep networks, unlike existing methods which depend on having a fully linear network or a linear decoder.

An important limitation of our unsupervised pathway attributions is that they do not always capture all pathways that differ significantly between two groups of interest, if those pathways do not represent major sources of expression variation (see Fig. 3b). We emphasize that our unsupervised attribution method measures a different quantity than supervised approaches, and should be considered a complementary approach rather than a replacement in applications where labeled data are present.

An assumption of our approach is that pathway databases provide good representations of the underlying biology of the system being modeled, which is an assumption shared by any method that incorporates pathway information into the modeling architecture. We note, however, that our incorporation of gene attributions specifically helps to alleviate this problem. Our gene attributions, like the pathway attributions, can be applied to arbitrarily deep networks, and allow for the biological interpretation of models with multiple latent variables per pathway. These attributions also allow the user to understand what biology is captured by the model in densely connected nodes that are not associated with a particular pathway.

Importantly, PAUSE was able to identify mitochondrial Complex I as a critical factor mediating A$\beta$-related pathophysiology in AD, pointing to the promise of mild inhibition of Complex I as a potential new paradigm for developing future AD therapeutics. This finding is supported by prior data indicating that mild inhibition of Complex I reduced A$\beta$ and tau levels in mouse models of familial AD [51], that diets rich in capsaicin — a natural product that can inhibit Complex I — are associated with lower serum A$\beta$ levels in the elderly [52], and that capsaicin reduced AD-associated changes in the hippocampus of rats with type 2 diabetes [53].

## Conclusions

We anticipate that our approach of combining principled attributions with interpretable model architectures will prove to be a broadly useful strategy in domains beyond gene expression. Rather than viewing attribution methods or biologically-meaningful representations as individual panaceas for machine learning interpretability, we hope that researchers will view these approaches as composable tools, each addressing unique needs. For example, principled feature attribution methods are currently applied to computer vision models, but most of these methods generate attributions at the level of pixels, which are not the most human-interpretable features and struggle to represent higher-level concepts [54, 55]. On the other hand, generative models like GANs can more clearly illustrate complex patterns differentiating domains, but lack the rigorous quantification of feature attribution methods [54, 56]. We believe future work that is able to rigorously quantify contributions of human-interpretable concepts in deep learning models will be useful in domains like microscopy, computer vision, and digital pathology.

Additionally, while biological pathways represent one useful representation, they will not necessarily be the only meaningful representation for all data types. For example, in the area of computer vision, researchers have proposed using additional labeled data to train medical image models with representations encoding high-level clinical

concepts, such as the presence or absence of bone spurs in x-ray images of knees for osteoarthritis severity determination [57]. Future work defining other useful conceptual representations will be helpful in biological applications, while finding methods such as the sparsely-constrained architectures used in this manuscript to learn representations without the need for costly additional labels will be useful in other domains.

Even in biology, pathway annotations may not always provide the most useful representations. In particular, translating interpretable approaches to multi-modal datasets will be an important future direction. Defining interpretable representations capable of grouping features across data modalities will be important to integrate these diverse datasets. For example, one could design a transcription factor-based latent space, linking both the expression measurements of genes annotated to be regulated by each transcription factor, and the chromatin accessibility of genome regions containing sequence motifs related to the binding of each transcription factor.

## Methods

### Model architectures

Biologically interpretable autoencoders are unsupervised deep learning models with latent variables corresponding to known biological pathways or other gene groupings. Methods to create biologically interpretable autoencoders can be grouped into two broad strategies. The first involves directly modifying the architecture of a network with sparse, masked connections. This approach is taken by methods such as VEGA (VAE Enhanced by Gene Annotations) [21], which masks the weights in the linear decoder of a variational autoencoder (VAE) so that each latent variable is only connected to genes in the pathway represented by that latent variable. The second involves a "soft" modification of the network weights using weight regularization during training. For example, the expiMap method (explainable programmable mapper), which is also a VAE with a linear decoder, allows all connections from all latent variables in its decoder to potentially have non-zero weight, but adds an L1 regularization penalty specifically to the weights corresponding to genes not present in the original pathway definitions [20].

In our experiments, we primarily used a *fully deep* architecture similar to that proposed in the pathway module variational autoencoder (pmVAE) approach [22]. This autoencoder network's encoder and decoder *both* have multiple non-linear layers. To better describe these networks, we can start with a description of a typical, unmodified VAE [58]. Standard VAEs are a pair of neural networks, an encoder with parameters $\theta$ and a decoder with parameters $\psi$, optimized to learn a low-dimensional distribution over latent variables **z** from high-dimensional data **x**, such as gene expression data. These networks are trained by stochastic gradient descent to maximize the variational lower bound on the likelihood of the data:

$$\log p_\theta(x) \geq \mathbb{E}_{q(z|x;\theta)}[\log p(x|z;\psi)] - \mathbb{KL}(q(z|x;\theta)||p(z)) = -\mathcal{L}_{\text{ELBO}}. \tag{1}$$

In order to learn a low-dimensional latent space **z** that corresponds to biological pathways, the pmVAE approach uses sparse masked weight matrices to separate the weights of the encoder and decoder neural networks into non-interacting modules for each pathway. These modules may have multiple hidden layers in both the encoder and decoder network, and may have multiple latent variables associated with each

pathway. The network's first layer is masked with a binary *assignment mask*, which ensures that each gene is only connected with non-zero weight to the hidden nodes of the modules corresponding to its pathways. Each subsequent layer is masked with a binary *separation mask*, which is a block diagonal matrix ensuring that non-zero connections only occur within pathway modules. In addition to the sparse masking modifications made to the encoder and decoder architectures, the pmVAE method also alters the training objective to add a *local reconstruction loss*:

$$\mathcal{L}_{\text{recon}}^{(\text{local})} = \frac{1}{K} \sum_p \frac{N}{N_p} \|\hat{x}^{(p)} - x^{(p)}\|_2^2, \tag{2}$$

where $\hat{x}^{(p)}$ is the reconstructed expression of the genes in pathway $(p)$, $x^{(p)}$ is the observed expression of the genes in pathway $(p)$, $\frac{N}{N_p}$ is a weighting term to weight each module's local reconstruction loss dependent on the number of genes in a particular module $N_p$ compared to the total number of genes $N$, and $K$ is the total number of modules considered.

In our experiments on bulk RNA-seq brain data, the brain gene expression samples were compiled from multiple data sources. In order to disentangle the batch effects, represented by a vector $c$, from the biological variation of interest, we wanted to train a *conditional* pathway module VAE (cpmVAE), and therefore maximize the *conditional* variational lower bound objective:

$$\log p_\theta(x|c) \geq \mathbb{E}_{q(z|x,c;\theta)}[\log p(x|c, z; \psi)] - \mathbb{KL}(q(z|x, c; \theta)||p(z|c)) = -\mathcal{L}_{\text{ELBO}}^c. \tag{3}$$

These condition labels are passed to each pathway module in the encoder, and again to each pathway module in the decoder. In order to ensure that the learned embedding is fully independent of the unwanted sources of variation encoded in the conditional labels, we add a regularization term to the loss based on the Hilbert Schmidt independence criterion (HSIC) between the latent embedding **z** and the dataset labels **c** [59, 60]. Therefore, our total loss function optimized for the cpmVAE architecture is:

$$\mathcal{L}_{\text{total}} = \mathcal{L}_{\text{ELBO}}^c + \mathcal{L}_{\text{recon}}^{(\text{local})} + \mathcal{L}_{\text{HSIC}}. \tag{4}$$

We implemented our cpmVAE model (and the pmVAE models) using the PyTorch deep learning library [61]. For both the cpmVAE and pmVAE models, we included BatchNorm1D layers following all but the final layers of both the encoder and decoder networks. We optimized the networks using an Adam optimizer with an initial learning rate of 0.001, and decreased the learning rate by a factor of 10 following each epoch if there was not a decrease in global reconstruction error on a held out validation set. All models were trained for 200 epochs, and the model checkpoint with the lowest reconstruction error on a held out validation set was used for downstream analysis. For single cell benchmark experiments, each pathway module had a hidden layer of 12 nodes, followed by a pathway latent space of 1 node. For all other experiments, each pathway module had a hidden layer of 12 nodes, followed by a pathway latent space of 4 nodes. For the cpmVAE experiments, the HSIC loss was multiplied by a factor of $1 \times 10^6$, to ensure this term was around equal magnitude with the other loss terms.

For our experiments involving models with linear decoders (see Additional file 1: Fig. S3), we used the Interpretable Autoencoder architecture proposed by Rybakov et al. [62]. This model is similar to the expiMap method [20], but is a deterministic autoencoder rather than a VAE. We trained these networks using the code from the intercode repository (https://github.com/theislab/intercode), and for both experiments the network was trained using the hyperparameters described in (https://github.com/theislab/intercode/blob/main/notebooks/intercode-api-Kang18.ipynb): 80 epochs with a batch size of 62 samples, a learning rate of 0.001, and the regularization hyperparameters of $\lambda_0 = 0.1$, $\lambda_1 = 0.93$, $\lambda_2 = 0.0$, and $\lambda_3 = 0.57$.

For the experiment demonstrating how PAUSE enables a similar unsupervised workflow between an unconstrained VAE and a classical PCA approach (see Additional file 1: Fig. S1), we used an autoencoder with 4 hidden layers in the encoder, 2000 in the first layer, 1000 in the second layer, 500 in the third layer, 250 in the fourth layer, then 10 nodes in the bottleneck dimension (each a normal distribution parameterized by a mean and a variance node, with backpropagation enabled using the reparameterization trick). The decoder network had the reverse structure, with a first hidden layer of 250 nodes, then 500, 1000, and 2000, before generating the full reconstruction. This model was trained for 20 epochs using an Adam optimizer with learning rate set to 0.001, and a batch size of 16.

For the results of our impute benchmark on a standard VAE (see Additional file 1: Fig. S4), we use the `VAEmodel` class found in the `models.py` of the linked github repository. The number of nodes in this model's layers is the same as in the pmVAE benchmark experiments. The pmVAE model is comprised of pathway modules, each with 12 nodes in a hidden layer followed 1 node in the latent space. If we call the number of pathways used to construct a pmVAE for a given dataset `num_pathways`, we construct the standard VAE with 12 x `num_pathways` nodes in a hidden layer and `num_pathways` in the latent space. The model was trained for 100 epochs using an Adam optimizer with a learning rate set to 0.001 and a batch size of 16.

### Attributions

Feature attribution is one of the largest and most well-studied classes of methods for machine learning interpretability. Methods in this class are based on concepts from cooperative game theory, like the Shapley value [29] and Aumann-Shapley value [30]. By framing the output of a complex, black box model as a game, and the input features as players in that game, these concepts can be used to understand which input features of a model most impacted that model's predictions. For example, the feature attribution method SHAP defines a cooperative game to be the expected value of a machine learning model evaluated on a sample of interest, conditional on the features present in each coalition, and then uses the Shapley value to allocate credit to those features [10]. Another feature attribution method, Integrated Gradients, calculates an Aumann-Shapley value by integrating the gradient of the model's output with respect to its input features, and can be used when machine learning models are differentiable, such as neural network models [12].

We propose a novel attribution method to understand which latent factors are important in unsupervised neural networks, based on Integrated Gradients [12] and

the Aumann-Shapley value [30]. However, rather than attributing the output of a supervised machine learning model to its input features, we attribute the reconstruction loss of an autoencoder model to its latent features. To make our formulation general enough to encompass both variational and standard autoencoders, we define an encoder network $f : \mathbb{R}^d \mapsto \mathbb{R}^h$ and a decoder network $g : \mathbb{R}^h \mapsto \mathbb{R}^d$. For a decoder, we can define the reconstruction error, $\ell(z) = \|x - g(z)\|_2^2$, as a function of a latent input $z$, which is either a sample drawn from the distribution parameterized by the encoder network given an input sample $x$ in the case of a variational autoencoder, or simply the deterministic embedding of the encoder network in the case of a standard autoencoder. To quantify the contribution of a particular latent node $i$ (which in the case of an interpretable autoencoder, corresponds to a biological pathway), we can therefore apply the following formula to get a local pathway importance value:

$$\phi_i^{\text{pathway}}(z) = (z_i - z_i') \times \int_{\alpha=0}^1 \frac{\partial \ell(z' + \alpha(z - z'))}{\partial z_i} d\alpha, \tag{5}$$

where $z$ is the value of the learned pathway embedding for a particular sample, $z'$ is a baseline value for that embedding, such as a vector of all 0s or the mean of that embedding's value over the dataset, and $\alpha$ is a scalar representing the distance on the straight-line path being integrated between the baseline and the actual value of the sample.

These local attributions have a variety of desirable properties [12], such as *completeness*, meaning that the attributions for each pathway node sum to the difference between the reconstruction error of the model at the input $z$ (when the pathway information is "present") and the baseline $z'$ (when the pathway information is "absent"). Completeness is important in that it gives the attributions a natural scale: each pathway's attribution represents the amount of variance explained by the model for the original sample that can be credited to that pathway. These local attributions are also *implementation invariant*, meaning the attributions are always identical for two functionally equivalent networks. This is important, in that it ensures the attributions reflect real functional differences between networks, rather than relying on artifacts based on particularities of the model employed.

To go from local pathway importance (the contribution of a particular pathway for a particular sample) to global pathway importance (the contribution of a pathway over an entire dataset), we can take the expected value of the local attributions over the samples in the original data:

$$\Phi_i^{\text{pathway}} = \mathbb{E}_{z \sim \mathcal{D}}[\phi_i^{\text{pathway}}(z)]. \tag{6}$$

Since these attributions can be calculated so efficiently, rather than sampling we calculate the local attributions for every point in the dataset. Because the local attributions allocate the reduction in the sample-level reconstruction error to each pathway, the global attributions allocate the reduction in the dataset-level reconstruction error to each pathway. Since the mean squared error is proportional to the variance in the original expression space explained by the model, allocating the reconstruction error to each pathway tells us how much variance is explained by each pathway.

We also note that our method is agnostic to the exact form of the loss used to train the model. For example, approaches like scVI model RNA-seq data as discrete counts under a negative binomial (or zero-inflated negative binomial) distribution by optimizing the decoder to parameterize one of these distribution types. For a model like this, rather than using the mean squared reconstruction error as the function $\ell(x)$ in the PAUSE definition, one could instead use the negative log likelihood of the observed data under the model. A notebook demonstrating the application of PAUSE to an autoencoder parameterized in this manner is available in the "Additional Notebooks" section of the Github repository.

For our gene-level attributions, we want to understand which genes are important contributors to each of the latent pathway nodes. For a particular node in the latent space $k$ that we wish to explain, we define the function $f_k(x) : \mathbb{R}^d \mapsto \mathbb{R}$ as the $k$th node in the encoder network's latent space. We can therefore quantify gene $j$'s contribution to pathway node $k$ by applying the Integrated Gradients formula to this function to get:

$$\phi_j^{\text{gene,k}}(x) = (x_j - x_j') \times \int_{\alpha=0}^{1} \frac{\partial f_k(x' + \alpha(x - x'))}{\partial x_j} d\alpha, \tag{7}$$

where $x$ is the observed expression for a particular sample, $x'$ is a baseline value for gene expression, such as a vector of all 0s or the average gene expression over the dataset, and $\alpha$ is a scalar representing the distance on the straight-line path being integrated between the baseline and the actual value of the sample.

To summarize local gene attributions into global gene attributions, we take the average of the magnitude of the local attributions over the samples in the original data:

$$\Phi_j^{\text{gene,k}} = \mathbb{E}_{x \sim \mathcal{D}}[|\phi_j^{\text{gene,k}}(x)|]. \tag{8}$$

It is necessary to average the magnitudes rather than the raw values for gene attributions to avoid cancelation effects from attributions with different signs [31]. This was not a problem for pathway attributions, as all attributions should have the same sign, as adding more pathway information should never *increase* the loss.

All attributions were calculated using the Path Explain repository (https://github.com/suinleelab/path_explain), which is a Python library for explaining feature importances and feature interactions in deep neural networks using path attribution methods [63–65]. All attributions were generated using the "attributions" method of the PyTorch explainer; the argument "num_samples" was set to 200, while the argument "use_expectation" was set to "FALSE."

### Pathway gene set definition

We use pathway modules defined by gene set annotations from the Reactome database for all single cell experiments [28]. This database contains a total of 674 gene sets (with a median gene set size of 27 genes). The number of gene sets included for a given model depends on the dataset. For all single cell datasets, we only include gene sets that contain a minimum of 13 genes from the given dataset. This yields a total of 200 gene sets for the PBMC INF-$\beta$ dataset, 129 gene sets for the intestinal epithelium dataset, 341 gene sets for the Jurkat anti-CD3/anti-CD28 dataset, 170 gene sets for the cancer cell lines

dataset, 242 gene sets for the K562 CRISPR perturbation dataset, and 203 gene sets for the BMMC AML dataset.

For the bulk Alzheimer's brain expression expermients, the pathway module architecture of the networks was defined using the Hallmark gene sets from MSigDB. These pathways are highly curated to "summarize and represent specific well-defined biological states or processes and display coherent expression" [66].

### Single cell expression datasets

#### PBMC INF-$\beta$

This dataset, from Kang et al. [32], contains human peripheral blood mononuclear cells (PBMCs) from eight patients with Lupus who were either treated with INF-$\beta$ or left untreated as a control. We followed the same preprocessing steps taken in prior analysis of this dataset [22, 62], which consists of library size normalization, removal of low variance genes and log transformation. After preprocessing, the final dataset we use contains 13,576 samples with 979 genes. The dataset contains 6359 control and 7217 stimulated cells.

#### Intestinal epithelium

This dataset, from Haber et al. [33], profiles the response of mouse small intestinal epithelial cells to pathogen exposures, specifically *Salmonella enterica* and *H. polygyrus.* Here, we include healthy control cells and cells 10 days after being infected with *H. polygyrus.* Preprocessing follows the same steps outlined by Weinberger and Lin [67] and involves retaining only the top 2000 most highly variable genes followed by library size normalization and log transformation. After preprocessing, the final dataset we use contains 5951 samples with 2000 genes. The dataset contains 3240 control and 2711 *H.polygyrus*-stimulated cells.

#### Jurkat anti-CD3/anti-CD28

This dataset, from Datlinger et al. [23], contains human Jurkat cells (immortalized T-lymphoctyes) that were either starved or stimulated with anti-CD3 and anti-CD28 antibodies. Preprocessing follows the approach described in Gut et al. [22], who filter out genes and cells with low expression, perform library size normalization and log transformation, and filter out low variance genes. After preprocessing, the final dataset we use contains 1288 samples with 2139 genes. This dataset contains 607 control and 681 stimulated cells.

#### Cancer cell lines

This dataset, from McFarland et al. [68], contains the transcriptional response of cells from 24 cancer lines to idasanutlin, a negative regulator of the tumor suppressor p53. Some cell lines contain wild type TP53, while others contain inactive mutant TP53. Preprocessing of this dataset follows the same procedure outlined by Weinberger and Lin [67] and involves retaining only the top 2000 most highly variable genes, library size normalization, and log transformation. After preprocessing, the final dataset we use contains 3097 samples with 2000 genes. This dataset contains 671 wild type samples and 2426 mutant TP53 samples.

### K562 CRISPR perturbation

This dataset, from Norman et al. [69], measures the transcriptional response of K562 cells to CRISPR perturbations of single genes and pairs of genes. The authors annotate clusters of cells according to transcriptional response. Here, we analyze unperturbed control cells and cells annotated as belonging to the "granuloctye/apoptosis" cluster. Pre-processing of this dataset follows the same steps outlined by Weinberger and Lin [67] and involves retaining only the top 2000 most highly variable genes, library size normalization, and log transformation. After preprocessing, the final dataset we use contains 11,895 samples with 2000 genes. This dataset contains 7275 control and 4620 perturbed cells.

### BMMC AML

This dataset, from Zheng et al. [70], contains single cell RNA-seq values of bone marrow mononuclear cells (BMMCs) of patients with acute myeloid leukemia (AML), taken before a stem cell transplant. The dataset also contains measurements from healthy control samples. Preprocessing of this dataset follows the same steps from Weinberger and Lin [67] and involves retaining only the top 2000 most highly variable genes, library size normalization, and log transformation. After preprocessing, the final dataset we use contains 11,982 samples with 2000 genes. This dataset contains 4457 healthy samples and 7525 pre-transplant AML samples.

### Bulk brain expression datasets

For experiments with AD, we used data from the ROSMAP, ACT, HBTRC, MAYO, and MSBB studies. Around half of the people in each cohort had been diagnosed with dementia by the time of death. ROSMAP RNA-Seq data and MSBB RNA-Seq data were made available by Sage Bionetworks on the AMP-AD Knowledge Portal with Synapse IDs syn3505732 and syn7391833, respectively. The ACT RNA-Seq data20 was collected by the Allen Institute for Brain Science, Kaiser Permanente Washington Health Research Institute (KPWHRI), and the University of Washington (UW), and it was made available with Synapse ID syn5759376. Mayo Clinic Brain Bank data was made available at Synapse ID (syn5550404; https://doi.org/10.1038/sdata.2016.89), and the Harvard Brain Tissue Resource Center (HBTRC) study was made available at the following Synapse ID (syn3159435). In each study, we used the protein-coding genes that have a nonzero RNA-Seq read count in at least one-third of the samples. Overlapping these genes across the three studies resulted in 16,252 genes which we used in our experiments. We used normalized and log-transformed RNA-Seq read counts for all datasets. For the subset of samples with neuropathology data, MSBB neuropathology data was made available by the AMP-AD Knowledge Portal of Sage Bionetworks through synapse.org with Synapse ID syn6101474. We accessed neuropathology data from the ROSMAP and ACT studies through data use agreements.

### Benchmark experiments

To benchmark how well our PAUSE pathway attributions identified important factors of latent variation explaining the observed gene expression measurements, we adapted two benchmarks of feature importance from the feature attribution literature [10]. The first

benchmark, our *impute* benchmark, measures how much the reconstruction error of a model increases when the pathways identified as important by an attribution measure are removed from a model. After training a pmVAE model [22] (12 nodes in a single hidden layer, and a single latent node for each module) on each of the 3 single cell RNA-seq datasets, pathways were ranked by one of five methods: (1) PAUSE attributions; (2) LR score, the accuracy of a logistic regression model trained to classify the ground truth perturbation using the latent pathway information [22, 62]; (3) KL Divergence between the learned posterior distribution and the isotropic gaussian prior; (4) random; and (5) the magnitude of the learned latent space scale parameter for each pathway. After attaining a pathway ranking for each method, pathway information is removed by masking the real latent embeddings with a constant value. The increase in reconstruction error over the dataset is then measured after each pathway is removed. Attribution methods that do a better job of identifying important pathways will increase the reconstruction error more quickly, leading to a larger area under the curve. These curves were summarized using the scikit-learn metrics auc function [71]. The second benchmark, which we termed a *retrain* benchmark, follows the same initial steps as the impute benchmark (train models and rank pathways), but then trains new models using only modules corresponding to the top pathways. In this benchmark, methods that perform better will decrease the reconstruction error more quickly as pathway modules are added. Finally, to benchmark the standard, unconstrained VAE models and the models with linear decoders, we followed the same steps as the impute benchmark, but trained the intercode models and VAE models as described above in the section on Model Architectures, rather than pmVAE models.

### Computational efficiency

We have recorded the compute time and resources needed for our analyses. We will focus our compute time analyses on a typical dataset used in these analyses, in this case, the Zheng et al. BMMC dataset [70] where we look at 11,982 samples, each with measurements for 1907 genes. In this case, we include 203 pathways in our sparse autoencoder. For model training, we use one NVIDIA RTX A4000 GPU. The mean train time over 10 random initializations, with max_epochs set to 100, was 13.3 min. PAUSE attributions were calculated using the Path Explain repository (https://github.com/suinleelab/path_explain), which is a Python library for explaining feature importances and feature interactions in deep neural networks using path attribution methods. All attributions were generated using the "attributions" method of the PyTorch explainer; the argument "num_samples" was set to 200, while the argument "use_expectation" was set to "FALSE. With this experimental setup, the mean time to compute PAUSE attributions was 22.54 s. In contrast, it took an average of 1.34 s to get supervised attributions using logistic regression.

### Sanity checks for bulk expression cpmVAE model

Before interpreting the learned embedding for our conditional pathway module VAE (cpmVAE) model trained on the AD dataset, we wanted to perform several sanity checks to ensure the model was sufficiently plausible and reliable. We first wanted to ensure that our proposed architecture (cpmVAE) was able to reconstruct the data with at least the

same accuracy as the previously proposed, non-conditional pmVAE model. We therefore randomly split the data into a train set (75% of samples) and a test set (25% of samples), trained both a pmVAE and a cpmVAE on the training set for 50 epochs, and measured the reconstruction error on the held out test set for each model. We repeated this procedure a total of 10 times with different random train/test splits, and compared the distribution of test reconstruction errors for the two models (Additional file 1: Fig. S12a). We found that the distribution of test reconstruction errors attained with the cpmVAE model was lower than the distribution of test reconstruction errors attained by the previously proposed pmVAE model (Wilcoxon rank-sums test statistic $= 1.96$, $p = 0.049$).

Next, we wanted to ensure that the dataset integration was successful, and that the conditioning and HSIC regularization indeed led to a pathway embedding that encoded less information about the data sources than the standard pmVAE embedding. To measure how clustered cells were according to dataset source in the latent space, we first embedded the pathway latent spaces of the cpmVAE model and the pmVAE model into 2 dimensions using the scikit-learn implementation of TSNE with the perplexity parameter set to 40.0 [71, 72]. We then applied k-means clustering to the latent space (with k set to the number of true data sources), and computed the adjusted Rand index (ARI) of the clustering [73]. When we compare the distribution of ARIs for the pmVAE embedding to the distribution of ARIs for the cpmVAE embedding (Additional file 1: Fig. S12b), we see that the embedding learned by the cpmVAE model has significantly less clustering according to dataset source (Wilcoxon rank-sums test statistic $= 3.78$, $p = 1.57 \times 10^{-4}$).

After demonstrating that the cpmVAE model removed unwanted sources of variation from its pathway embedding, we wanted to ensure that our cpmVAE embedding still retained biological information. For a small subset of the brain expression samples (664 samples), associated amyloid-$\beta$ protein density measurements via immunohistochemistry (A$\beta$ IHC) were available. To ensure that biological information related to neuropathological phenotype was not being lost by our model, we wanted to measure how well A$\beta$ IHC could be predicted from the latent space, as compared to the original full transcriptomic representation. Our hypothesis was that if the biologically relevant information in the latent space was being preserved, prediction of A$\beta$ density should be no worse when using the cpmVAE embedding than when using the original gene space embedding. For 20 random train/test splits of the data, we compared the Spearman correlation between predicted and actual A$\beta$ density for linear models trained on the full gene expression measurements to the Spearman correlation between predicted and actual A$\beta$ density for linear models trained only on the cpmVAE pathway embeddings (Additional file 1: Fig. S12c). All linear models were trained using the scikit-learn RidgeCV function [71], with a hyperparameter search for alpha values ranging from 1e−3 to 1e−2. We see that the cpmVAE embedding features actually lead to significantly more predictive models of A$\beta$ density than the original gene expression features (Wilcoxon rank-sums test statistic $= 2.81$, $p = 4.90 \times 10^{-3}$).

While these results are certainly suggestive of an improvement in the control of nuisance variables and an improvement in one biological signal represented, they also do not necessarily demonstrate that all of the nuisance variation has been removed, or that all of the biological signal is optimally preserved, as there are likely signals in the dataset from biological processes not related to density of A$\beta$, and the preservation of these

signals is not explicitly assessed here. Prior work has been done on the problem of how much correction should be applied when trying to extract biological signatures from gene expression compendia [74]. The simulations in this paper show that choices surrounding correction depend upon the number of sources of variation being controlled, the size of the dataset in question, and the magnitude and structure of the signals to be extracted. Knowing information about the magnitude and structure of the signals of interest in real-world, non-synthetic datasets is almost always not possible, and would serve as interesting future work to help guide modeling decisions about nuisance factor correction.

Finally, to compare the overlap between supervised approaches and pathway attributions from PAUSE on the AD dataset, we generated a ranked list of pathways using the supervised LR score method for the 664 brain expression samples with associated AB IHC protein density measurements (see Additional file 2: Table S1). We can see that the pathways identified by the LR score, in this case, do significantly overlap with the top pathways identified by our unsupervised attributions. For example, both supervised and unsupervised attributions identify pathways related to sources of oxidative stress: the Reactive Oxygen Species pathway and the Peroxisome pathway for the supervised attributions, and the Mitochondrial Oxidative Phosphorylation pathway for the unsupervised attributions. Not only are these pathways all related to potential sources of oxidative stress, but they also share overlapping genes. In particular, genes encoding different subunits of the NADH:ubiquinone oxidoreductase complex are present in both the Mitochondrial Oxidative Phosphorylation pathway and the Reactive Oxygen Species pathway. The supervised and unsupervised approaches also overlap in identifying TNF alpha signaling via NF-kB as an important pathway in this dataset, which is, again, a key pathway known to mediate neuroinflammation.

### Identification of *C. elegans* homologs

To enable biological testing of the human genes identified using our computational analysis, we obtained the Reciprocal Best Hits (RBHs) between human and *C. elegans*. We first identified all unique protein sequences for each potential marker gene using the biomaRt R package available on CRAN [75, 76]. Then, we used the NCBI BLAST tool to identify the *C. elegans* orthologs for each complete human protein query sequence [77, 78]. We downloaded the *C. elegans* protein sequences from wormbase.org/species/c_elegans (release WS266). We took into account only the protein pairs mapped from human to *C. elegans* with a BLAST *e*-value smaller than $10^{-30}$. For each *C. elegans* isoform, we identified the corresponding human genes, again using the NCBI BLAST tool, and used only the orthologs that achieved a BLAST *e*-value smaller than $10^{-30}$. This process resulted in high-confidence RBHs for us to test in *C. elegans*.

### *C. elegans* strain, cultivation, and RNAi treatment

Standard procedures for *C. elegans* strain maintenance and manipulation were used, as previously described [79, 80]. All experiments were performed using the GMC101 strain expressing the human A$\beta$1–42 peptide under the unc-54 promoter [49]. Experimental worm populations of GMC101 animals were obtained from the Caenorhabditis Genetics Center (CGC) and cultivated on NGM plates with OP50 *E. coli* at 15C [49, 81]. Care

was taken to ensure that the animals were never starved and the plates remained free of contamination.

The gene-specific RNAi clones were obtained from the commercial Ahringer or Vidal *C. elegans* RNAi-feeding libraries (BioScience, Nottingham, UK). Each bacterial clone was struck-out onto LB plates containing carbenicillin (50 μg/ml) and tetracycline (10 μg/ml). Single colonies were then seeded into 5 ml LB + carbenicillin (50 μg/ml) and tetracycline (10 μg/ml) for growth overnight on a 37 °C rotator. One hundred microliters of each overnight culture was then inoculated into 10 ml of LB containing carbenicillin (50 μg/ml) and tetracycline (10 μg/ml) and IPTG (5mM) and incubated on a 37 °C rotator for 4 h. Each bacterial growth was then centrifuged at 3500 X G for 25 min, decanted, and the pellet resuspended in 0.5 ml of LB containing carbenicillin (50 μg/ml), tetracycline (10 μg/ml), and IPTG (5 mM). To verify that the RNAi plasmid DNA contained the expected gene target, each RNAi clone was purified and assessed through PCR (polymerase chain reaction) with sequence-specific primers or through Sanger sequencing.

### Nematode paralysis assays

Paralysis assays were performed by visually inspecting recordings of the animals daily to determine if they were capable of normal locomotion or if they were paralyzed and unable to transit the agar plate. A custom-built robotic system, the WormBot [82], was equipped with a digital camera and was used to obtain images of individual wells of a 12-well plate at 10-min intervals over the entire course of the experiment. Each well contained 30–40 individual GMC101 animals expressing A$\beta$. Using a custom-built web interface that enabled manual annotation of serial images from each plate, the age at which each animal stopped moving could be easily determined. We applied this system to the transgenic A$\beta$ model line GMC101 to determine the time of paralysis onset for each individual animal across all RNAi experiments. Statistical significance of mean paralysis time-points between RNAi conditions was determined by a weighted log-rank test [83, 84]. All key results were independently verified using standard manual micro-dissection methodologies.

Prior to loading on the experimental plates, animal populations were propagated on high-growth plates seeded with NA22 *E. coli*. Worm populations were developmentally synchronized by hypochlorite treatment, and the remaining eggs were deposited on unseeded plates overnight. Synchronized larval stage 1 animals were washed off unseeded plates and moved onto standard *C. elegans* RNAi plates containing carbenicillin (50 mg/ml), tetracycline (10 mg/ml), and IPTG (5 mM) 48 h at 20 °C. These developmentally synchronized, late larval stage 4-populations were then washed and transferred to their respective RNAi conditions on 12-well plates. We used standard RNAi conditions plus FuDR (100 ug/ml) to prevent progeny and nystatin (200 mg/ml) to prevent fungal growth [79]. Each RNAi condition was tested in 2–3 wells as technical replicates. At least three biological replicates, each started on different weeks, were conducted for each RNAi clone. Animals were maintained at ambient room temperature (generally 22–24 °C) over the course of the paralysis assay.

## Supplementary information

---

**Additional file 1.** Figures S1-S12.

**Additional file 2.** Table S1 (ranked list of pathways for the 664 brain expression samples with associated AB IHC protein density measurements).

**Additional file 3.** Table S2 (list of all enriched pathways for the principal components and latent variables in panels c and d of Fig. S1).

**Additional file 4.** Review history.

---

### Acknowledgements

The results published here are in part based on data obtained from the AD Knowledge Portal (https://adknowledgeportal.org). Access to these datasets may be obtained after registering for a Synapse.org account, agreeing to acknowledge data used in any publications, and submitting a data-use certificate (separately as needed for each dataset). Our study uses the following datasets (with listed Synapse IDs; URLs): ACT (syn5759376), ROSMAP (syn3219045; https://doi.org/10.1038/s41593-018-0154-9), MSBB (syn3159438), Mayo Clinic Brain Bank (syn5550404; https://doi.org/10.1038/sdata.2016.89), and the Harvard Brain Tissue Resource Center (HBTRC) study (syn3159435).
Thank you to Dr. Alec Wilkens for helpful discussion about CRISPR-based assays. Thank you to Ethan Weinberger and Ian Covert for helpful discussion about generative deep learning models and feature attribution methods.
This work was funded by National Science Foundation [DBI-1759487, and DBI-1552309]; National Institutes of Health [R35 GM 128638, R01 NIA AG 061132, P30 AG 013280, 5-T32-AG-52354-5].

### Peer review information

### Review history

The review history is available as Additional file 4.

### Authors' contributions

JDJ, AS, SC, BWB, MK, and S-IL wrote the manuscript. JDJ coded the model architecture and training scripts. JDJ wrote the model explanation software. JDJ and AS ran computational experiments. JDJ, SC, AS, and S-IL designed computational experiments. BWB, JCR, TIL, and MK designed and ran *C. elegans* experiments. The authors read and approved the final manuscript.

### Authors' Twitter handles

Twitter handles: https://twitter.com/joejanizek (Joseph D Janizek).

### Code availability

Code to reproduce results and figures is available at: https://github.com/suinleelab/PAUSE.

### Funding

This work was funded by National Science Foundation [DBI-1759487, and DBI-1552309]; National Institutes of Health [R35 GM 128638, R01 NIA AG 061132, P30 AG 013280, 5-T32-AG-52354-5].

### Availability of data and materials

The datasets analyzed during the current study are all publicly available online from the Gene Expression Omnibus (GEO), Short Read Archive (SRA) or from the AD Knowledge Portal (https://adknowledgeportal.synapse.org/). Versions of the datasets as pre-processed in this manuscript are available in our Zenodo repository (10.5281/zenodo.7721287), in the zip directory "pause_sc_datasets." For full description of pre-processing of datasets, see Methods section above. Original repositories are listed as follows. Jurkat Anti-CD3/Anti-CD28 dataset is available at GEO Series GSE92872 [85]. PBMC dataset is available at GEO Series GSE96583 [86]. Intestinal epitelium dataset is available at GEO Series GSE92332 [87]. Bone marrow mononuclear cell (BMMC) dataset is available at Short Read Archive under accession number SRP073767 [70], and these data are also available at http://support.10xgenomics.com/single-cell/datasets. Cancer cell lines dataset is available through Figshare [88]. K562 perturbations dataset is available at GEO Series GSE133344 [89]. Bulk AD data was available in the AD Knowledge Portal (https://adknowledgeportal.org). Access to these datasets may be obtained after registering for a Synapse.org account, agreeing to acknowledge data used in any publications, and submitting a data-use certificate (separately as needed for each dataset). Our study uses the following datasets (with listed Synapse IDs; URLs): ACT (syn5759376), ROSMAP (syn3219045; https://doi.org/10.1038/s41593-018-0154-9), MSBB (syn3159438), Mayo Clinic Brain Bank (syn5550404; https://doi.org/10.1038/sdata.2016.89), and the Harvard Brain Tissue Resource Center (HBTRC) study (syn3159435).
 Code to reproduce results and figures is available at: https://github.com/suinleelab/PAUSE. A stable Zenodo repository is available under the following doi: 10.5281/zenodo.7721287 [90]. All code is open source, licensed under an MIT License.

## Declarations

### Competing interests

The authors declare that they have no competing interests.

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

## 