## [**Additional file 4.** Review history. · Genome Biology]

Review History

First round of review

Reviewer 1

Were you able to assess all statistics in the manuscript, including the appropriateness of statistical tests used? No.

Were you able to directly test the methods? No.

Comments to author:

The authors claimed that they proposed a novel unsupervised pathway attribution method to better identify major sources of transcriptomic variation than prior methods. However, the main idea of their method is actually borrowed, if I cannot use "copied", from another work called pmVAE (<https://doi.org/10.1101/2021.01.28.428664>), which is also cited in this manuscript as ref. [22].

1. Honestly, I don't see any novelty or additional contributions of this work over the existing pmVAE work. The only difference I could see is the authors of the current manuscript described the contribution in a different way as "principled feature attributions" (either pathway level or gene level), rather than "interpretable representations". However, they are exactly the same for gene expression data analysis.

2. although the current manuscript is of the exactly same idea as the pmVAE work, the authors did not give a clear description and illustration as the pmVAE did. For example, figure 1 doesn't help the reader at all to understand the key idea of their methods and what is exactly the architecture of their autoencoder was also not provided in the method sections.

3. The author only did very few tests on limited datasets. They should give more applications of their method in different scenarios to demonstrate its significance in the involved area.

4. As a method or model work, it's unacceptable that no source codes are provided for others to use and to reproduce their results.

5. Additional minor comment: Figures 5a and 5b are not cited in order on page 13, where Figure 5b was cited first in the text.

Reviewer 2

Were you able to assess all statistics in the manuscript, including the appropriateness of statistical tests used? Yes: The statistical tests used are appropriate.

Were you able to directly test the methods? No.

Comments to author:

I recommend accepting this manuscript with minor revisions as suggested below.

In this manuscript, the authors have developed a pathway attribution method PAUSE that utilizes unsupervised analysis of gene expression data, along with pathway representations leading to more pathway-based biological interpretations than existing methods. In this manuscript, the authors have developed a pathway attribution method PAUSE that utilizes unsupervised analysis,

along with pathway representations leading to more pathway-based biological interpretations of gene expression data. The strength of the method lies in pathway and gene level attributions that have been demonstrated to identify sources of transcriptomic variation and increase interpretability of the models respectively. The authors have demonstrated the utility of this method in identifying known and novel pathways showing most variation in 3 scRNA-Seq datasets with and without labeled data, as well identifying, and validating pathway and genes in a bulk RNA-Seq dataset.

Comments to author (in order of priority)

1. The paper emphasizes that PAUSE should complement supervised approaches for pathway identification. However, from Fig 3, the number of pathways unique to supervised approaches are limited. What is the justification for using these tools complementarily? It would be useful to have recommendations on how supervised approaches could be used in conjunction to PAUSE.
2. Please specify the link to the git repo for the method in the abstract.
3. In section 2.5, the authors have accounted for batch effects from data source by modifying the pmVAE architecture. Based on Fig 5c, although a lot of the data source variation is corrected for, there still appear to be distinct clusters by batches. Please address how cpmVAE can address multiple known sources of variation without over-correcting for transcriptomic variation.
4. Considering that the overlap between
5. Considering the limited overlap between supervised approaches and pathway attributions from PAUSE in sc datasets, how would those compare for AD bulk RNA-Seq data? Would you expect neuronal, neuroinflammatory pathways to be detected with supervised approaches?
6. In section 2.4, you demonstrate that the densely connected unannotated nodes represent genes relevant to the ground truth for the perturbation of PBMCs with IFN- β . Please explain- 'we omitted particular pathways that should have been most relevant for the perturbation'. How were these pathways selected? Generally, for a perturbation, there is a cascade of downstream changes likely to occur. So, just removing IFN pathway may not be sufficient to show that the unannotated nodes represent ground truth genes- Supp Fig 3.
7. The computational usability of this method warrants more discussion- compute resources needed, compute time benchmarking etc.
8. The implementation of the PAUSE method also requires further discussion- 1) for the method to run effectively, what sample size of sc or bulk datasets are needed; 2) how do you recommend handling preprocessing and normalization of RNA-Seq data; 3) what method of imputation do you recommend for missing values; 4) Is this method fairly translatable to other omics datasets or integrative datasets etc.
9. In Fig 4, how are the 'most important' genes from each node defined- please mention this briefly in the figure legend.

Reviewer 3

Were you able to assess all statistics in the manuscript, including the appropriateness of statistical tests used? Yes.

Were you able to directly test the methods? No.

Comments to author:

In this paper, the authors develop a method PAUSE that enables one to interpret unsupervised deep learning models, primarily autoencoders, used to model gene expression data from bulk and single cell assays. PAUSE builds on top of an existing framework called pathway module VAE (pmVAE) and is extended to incorporate also batch effects and batch removal in a conditional pmVAE (cpmVAE). The pmVAE model incorporates known gene-pathway memberships while learning the underlying latent space. PAUSE uses a new feature attribution method that identifies pathways that are most useful for explaining the overall variation in the data, as well as individual genes in each pathway that are the major contributors to the overall expression variation. PAUSE is applied to both single cell datasets with known class labels as well as bulk dataset with known batch effects. PAUSE was able to find the major sources of variation and is able to better rank potential pathways (as assessed by the impute and retrain metrics) from single cell data. On bulk data for various AD patient samples PAUSE identifying a mitochondria related pathway which was validated using a *C. elegans* model. The paper is well written and does a very nice job of contextualizing their work in light of the current literature. The paper also addresses an increasingly important problem of interpretability of ML models. I have a few questions/comments about the advantage of this approach over the pmVAE approach and simpler approaches and some claims in the paper that need additional clarification with some experiments.

1. Although the paper is well written it is initially a bit misleading as to what the proposed method is doing. Initially I thought the authors are developing a new unsupervised learning method for modeling gene expression data which has the automatic ability to also find informative pathways and that might not need the information of existing pathways. However, what the contribution of this paper is really a new feature attribution method which builds on top of an existing framework. This part of the paper needs to be better described. I feel the "Overview of PAUSE" section right before the results should talk in more detail about the pmVAE and cpmVAE models. Even the methods don't really go into much details about the pmVAE model. To understand what PAUSE is doing, we need to understand pmVAE and its various model components. It would be helpful to describe this better.
2. The comparison of the specific pathways that are identified using the PAUSE versus supervised methods could be more compelling. While the authors discuss the pathways not identified by PAUSE, what would be even more compelling is if the authors could discuss pathways that only PAUSE identifies and that these are indeed relevant to what the supervised methods are giving us.
3. The gene attribution aspect of PAUSE helps to further add interpretability, but I feel this section could be better described and have more results. First, it would be helpful to more clearly describe perhaps upfront what is a node in a pathway. Is this a latent factor or is it a gene in the pathway. The model is essentially based on pmVAE so it would be very helpful to describe the pmVAE model in more detail and what nodes really correspond to in the model. It is also not clear what the authors meant by "removing a pathway". Does it mean it was removed from the input/architecture of the VAE? Where exactly is a dense node added in the model?
4. Related to 3., the finding of the correlated and uncorrelated gene pairs is interesting, but this is

shown only for one pair of genes. Fig 4D is a very common scatter plot for two genes that are somewhat correlated. I am not sure how we can interpret the specific cell populations that exhibit correlation vs not. Is it possible to identify these populations from the latent factors? Overall, the results here felt a bit anecdotal and more comprehensive analysis of gene attribution could be done.

5. The authors say that the approach is broadly applicable to any unsupervised (auto-encoder)-based method. However, PAUSE is demonstrated primarily with pmVAE and it is not clear how well PAUSE would work for VAEs with less structure or prior knowledge. Some more experiments could demonstrate the generality of PAUSE. For example, one could use more noisy or incomplete pathway information and see how much PAUSE's prioritization schemes are affected. It would be nice to show PAUSE with other types of VAEs as well.

6. The authors applied their method to an AD expression dataset and identified Oxidative phosphorylation and several mitochondrial genes that might be involved. A *C. elegans* model of AD was used to show that these genes have a phenotype when knocked down. These experiments could be described in detail. It was not clear based on the ranking of the genes, why one would see the increase in time for paralysis onset. What does this mean in terms of the role of the mitochondria in AD? Are these genes overexpressed in some way in AD patients?

7. I could not find a dedicated codebase for PAUSE and the cpmVAE model. The linked page https://github.com/suinleelab/path_explain seems to be associated with an already published paper. This needs to be clarified.

Dear Dr. Pang,

Thank you for considering our manuscript “Principled feature attribution for unsupervised gene expression analysis” for review by *Genome Biology* and for allowing us to submit a point-by-point response and incorporate the comments that we have received into a new revision of the manuscript.

In regards to the two points related to the necessary formatting for a ‘Method’ article that you referenced in your original email, we have (i) shortened the abstract so that it is under 100 words, (ii) updated the manuscript with links to source code available under an open source MIT license on Github (<https://github.com/suinleelab/PAUSE>), and (iii) uploaded the version of the code used in the paper to a Zenodo repository (doi:10.5281/zenodo.7331673). Accession information for the Github and Zenodo repositories are now listed in the Availability of Data and Materials section of the manuscript.

Thank you very much for submitting your manuscript to Genome Biology, and please accept my apologies for the delay in replying to you about it. It has now been seen by three referees and their comments are accessible below. As you will see from the reports, the referees find the manuscript of potential interest, but they raise serious concerns that additional demonstration and experiments are needed to show the advance of the approach. In particular, Reviewers 1 and 3 have concerns about the novelty and advance over pmVAE. They also ask for further demonstration and application to demonstrate the broad applicability. Please also ensure that the code is made available to reviewers. It seems to us to be essential that all of the referees’ concerns are fully addressed, in the form of a revised manuscript, before we can reach a final decision on publication.

We would also like to thank the reviewers for their careful consideration of this manuscript and their many suggestions for improvement. In response to the reviewers’ comments, we have made major changes that we feel substantially improve the manuscript and address the reviewers’ concerns, in particular the concerns you highlighted regarding the novelty and advance of our approach, and further demonstration of its broad applicability.

Here is a summary of the changes (followed by a full point-by-point response below):

1. Clearer description of our method and demonstration of novelty and advances

Both reviewers 1 and 3 mentioned that the main contribution of our manuscript was unclear. As put by R3, “although the paper is well written it is initially a bit misleading as to what the proposed method is doing.” To clarify our contributions, we have completely redesigned our concept figure in Fig. 1 and re-written the “Background” section as well as the section on an

“Overview of PAUSE approach.” We believe these changes make it more clear that our approach is a general unsupervised workflow that combines (1) learning a latent space with an autoencoder, (2) ranking the important latent variables using the novel unsupervised *attribution method* proposed in the paper, and (3) interpreting the biological significance of the important latent variables. While the biologically-constrained architectures used for Step (1) in this workflow are similar to those proposed previously, our updated manuscript now makes it more clear that it is the novel attribution method in step (2) and *not* the architecture that is the primary contribution. We also, however, have added new experiments demonstrating the importance of our proposed conditional architecture modifications in step 1 (see summary point 4 below).

2. Demonstration of applicability of unsupervised attributions to more models.

One point made by all 3 reviewers was related to the broad applicability of our method. While we had initially demonstrated the utility of our unsupervised attributions for ranking the latent variables of a biologically-constrained autoencoder trained with a mean squared reconstruction error, in this revision we have added several experiments to show that our unsupervised attributions can be useful in more cases. We now show that our unsupervised attributions can be used to analyze the important factors learned by standard, unconstrained VAEs through (a) ablation experiments that validate that our attributions identify the most important factors in these unconstrained architectures, and (b) an experiment that shows that our attributions enable a classic general workflow using deep autoencoders. We also show that these unsupervised attributions are agnostic to the specific loss used in the autoencoder, and (c) add code that demonstrates that our attributions can also be applied to discrete autoencoders trained with a negative binomial loss (scVI-type models). In conclusion, we show that our attributions can be broadly useful across many types of models.

3. Demonstration of applicability of PAUSE analysis to more datasets.

Another point made by reviewer 1 was that the number of datasets and experiments on these datasets in our initial manuscript was “limited.” We have added 3 new RNA-seq datasets to our analysis and analyzed the top pathways found by PAUSE attributions in these new datasets, further showing the utility of this method for understanding biological data across experimental settings. We also ran our benchmarking experiment across 20 different train/test splits of each of these 3 datasets to validate that our attributions are identifying the correct top pathways in these datasets as well, amounting to a comparison of our method to 4 alternative approaches across *60 new experiments*.

4. Demonstration of importance of conditional architecture modification.

While we hope that the improved version of our manuscript emphasizes that our primary methodological contribution is a novel unsupervised feature attribution method (to identify the

most important latent variables learned by a deep autoencoder), we have also added another experiment to show the importance of the architectural improvement we have made to the pmVAE architecture with our cpmVAE architecture. Working with a newly added dataset of single cell RNA-seq data of a variety of cell lines treated with the MDM2 antagonist idasanutlin, we demonstrate how conditioning on cell type allows our unsupervised attributions to correctly identify the cell cycle-related expression changes mediated by MDM2 and TP53 caused by idasanutlin in this dataset, rather than cell line-specific differences in expression. This experiment also highlights the difference between unsupervised and supervised attributions, which ask fundamentally different questions of the data, and consequently yield different answers. In the unconditioned model, the unsupervised attributions *correctly* identify that the primary source of variation in the dataset is the heterogeneity in cell types. After conditioning on cell type, the unsupervised attributions correctly identify that the primary source of heterogeneity in the dataset is cell cycle variation, due to the effects of MDM2 antagonist idasanutlin. Supervised attributions always identify the expression signal linked to a particular label, whether or not that label is the primary biological signal in that dataset. Sometimes this may be desirable, but it is not always the case that this is the most interesting signal.

Thanks to the recommendations from both reviewers and editors, we believe that our revised manuscript is much stronger, and we hope that you will find it suitable for publication in *Genome Biology*. We are grateful for your valuable insight and for your consideration of our work.

Sincerely yours,

Prof. Su-In Lee, PhD

Below is a point-by-point response to the referee comments. The reviewer's comments are in blue and italics, and our replies are in black.

Reviewer #1: The authors claimed that they proposed a novel unsupervised pathway attribution method to better identify major sources of transcriptomic variation than prior methods. However, the main idea of their method is actually borrowed, if I cannot use "copied", from another work called pmVAE (<https://doi.org/10.1101/2021.01.28.428664> [doi.org]), which is also cited in this manuscript as ref. [22].

1. Honestly, I don't see any novelty or additional contributions of this work over the existing pmVAE work. The only difference I could see is the authors of the current manuscript

described the contribution in a different way as "principled feature attributions" (either pathway level or gene level), rather than "interpretable representations". However, they are exactly the same for gene expression data analysis.

We thank the reviewer for pointing out that the initial description of our contributions in our manuscript was not clear, and we have edited the text of the manuscript and *completely redesigned* the concept figure (Fig. 1) to clarify the novelty of our approach.

From a machine learning perspective, the primary novelty of our paper is a new feature attribution method. This method, which is referred to in the paper as our “pathway attribution” method or “pathway loss attribution,” enables the *unsupervised* calculation of the importance of the latent factors of any neural autoencoder model (including biologically-constrained models like pmVAE, VEGA, and Interpretable AE, and even standard, unconstrained VAEs). Our attribution method is a new approach in a class of methods like SHAP, Integrated Gradients, and DeepLift. Our attribution differs from these methods, and is unique in that it applies to unsupervised machine learning models, allowing us to calculate the importance of latent variables learned by these models. It works by splitting the total variance explained by the model to each of the model’s latent variables, and is calculated by taking the integral of the gradients of the model’s reconstruction error with respect to its learned latent variables along a path from a neutral baseline to each observed point in the dataset (see new concept Fig. 1c).

The existing pmVAE work, in contrast, proposes a new architecture of autoencoder that uses sparsely connected modules so that the latent variables are constrained to only be a function of genes in a particular pathway, making these variables an interpretable representation in the sense that each latent variable corresponds to a pathway. Our primary contribution is ***a novel unsupervised pathway attribution method*** that allows the user to prioritize which of these pathways is the most important, by calculating how informative each pathway (more specifically, each sub-pathway node in the bottleneck layer) is for the reconstruction of the original expression. We demonstrate empirically (in Fig. 2) that our method prioritizes pathways more effectively than other possible approaches.

An intuitive way to understand our contribution can come from reading how unsupervised analysis of RNA-seq datasets was performed prior to deep learning. A great example is present in the paper proposing the Crop-seq technology, “Pooled CRISPR screening with single-cell transcriptome readout” (2017). In their Result section on *Single-cell CRISPR screening for T-cell-receptor induction* (page 299, column 2), they say “We established a transcriptome signature of TCR induction directly from the CROP-seq data by applying dimensionality reduction to single-cell RNA-seq profiles. Principal component analysis for cells grouped by gRNA target genes separated naive and TCR-induced Jurkat cells along the first principal component, which defined a TCR induction signature of 165 genes. This signature was enriched for genes known to be relevant in TCR signaling.” This same workflow is also employed in the 2015 paper by Macosko et al. proposing the Drop-seq technology, “Highly parallel genome-wide expression profiling of individual cells using nanoliter droplets” (see page 1204, bottom of column 2).

Essentially, their workflow follows these steps: (1) apply a dimensionality reduction method to factor the observed expression into latent variables presumably corresponding to biological signal; (2) rank these latent variables according to their importance; (3) interpret the biological meaning of the most important latent variables. In principal component analysis, the components are inherently ranked by the amount of variance explained by construction: the first PC is the coordinate that maximizes the variance explained when the data is projected onto that coordinate, the second PC is the coordinate that extracts the maximum variance from the data after the first PC has been subtracted, and so on. The importance of the latent variables is hard-coded into the algorithm. Deep learning-based variational autoencoders do not have a parameter or a value encoding any such importance. **Our primary contribution** in our manuscript, which draws on mathematics from the area of explainable AI, or more specifically, the feature attribution literature, is proposing **a way to calculate the importance of the latent variables of a deep learning model**.

We have added a new supplementary figure (see below) that demonstrates how our unsupervised pathway attribution method can be used in conjunction with a standard, unconstrained autoencoder to identify the most important latent variables in this model, then use gene attributions to find the enrichments. We first apply both PCA (scikit-learn implementation, with `n_components = 10`), and a VAE (10 latent nodes in the bottleneck layer), to the Jurkat T cell activation dataset. After training both of these models, we want to know which of the latent nodes in the bottleneck layer or principal components were most important. We can use the “`explained_variance_`” method from the scikit-learn implementation of PCA to calculate the contribution of each PC to the reconstruction. Our PAUSE pathway attributions can then provide the analogous calculation for the VAE model. Comparing these attributions between PCA and the VAE reveals interesting differences between the models learned. For example, the VAE learns a sparser model than PCA, representing a greater proportion of the observed gene expression variation in a smaller number of latent variables (see panels a and b in new Supplementary Fig. 1, included below, variance explained by latent variables from PCA on top, variance explained by latent variables from VAE on bottom).

We can then create a scatter plot of the learned embedding for the two most important PCs or LVs, and use gene attributions to interpret the biological significance of these dimensions (see panels c/d). We generate the top genes for each PC (using the magnitude of the gene loadings) and the top genes for each VAE LV using our integrated gradients-based gene attribution method (average magnitude IG attributions for the learned mean of the LV averaged over all cells in the dataset). We then take the top 100 genes and check for enrichments in biological processes using the enrichment testing in the web tool StringDB (see Supplementary Table 2 for exhaustive list of pathway enrichments found). We find that the top PC is mostly enriched for cell cycle-related genes, while the second PC is enriched for both cell-cycle related genes but also T cell activation pathway genes. Interestingly, for both of the top VAE LVs, there are many enrichments for both cell cycle processes AND T cell activation processes.

These results highlight a difference between biologically-constrained models of gene expression and unconstrained models. The PCs and latent variables in unconstrained models represent the expression of multiple biological processes simultaneously (rather than a single pathway in a

constrained model), possibly indicating a common regulation of these processes, or a hierarchical relationship between them (e.g. downstream effects of one process or pathway on another). These results also highlight the utility of pathway and gene attributions for unsupervised analysis of gene expression. The type of workflow done with PCA in the past (learn PCs/LVs, find the important ones, identify which genes contribute to these PCs) was not possible without a method like PAUSE, which is needed to identify the important latent variables for an unsupervised model.

We have revised the Background section of the manuscript to clarify where our approach fits in this workflow, see manuscript lines 52-60 (Manuscript changes are in bold):

Our paper aims to demonstrate that these two trends in biological interpretability are not mutually exclusive, and that principled attribution methods can improve the analysis of unsupervised models of gene expression analysis by quantifying the importance of pathways. **We first outline a general workflow for unsupervised analysis of gene expression data, comparing classical linear approaches like principal component analysis (PCA) to more contemporary deep learning-based approaches. This outline allows us to identify a step in the workflow that has been neglected by previous methods.** We then propose a novel, fully-unsupervised attribution method and demonstrate how it can be used to identify important pathways in transcriptomic datasets when combined with biologically-constrained autoencoders. This allows for fully unsupervised analysis of gene expression data. We next show how existing, feature-level attribution approaches still provide useful information for annotated, unsupervised models.

We have also added significant text to the revised manuscript in the first results section to more clearly describe our contribution, see manuscript lines 65-96 (Manuscript changes are in bold):

To understand how game-theoretic attributions can improve the unsupervised analysis of gene expression data, it is first helpful to understand a representative unsupervised workflow (Figure 1a) [23,24]. In the past, researchers have used linear approaches like Principal Components Analysis (PCA) to (1) learn a low-dimensional representation of gene expression data, (2) rank the latent dimensions according to the amount of variance in the original data explained by each dimension, and finally (3) interpret the biological meaning of the most important dimensions. Finding the importance of each latent dimension is straightforward in PCA, as the coordinates are arranged in descending order of variance in the data explained by construction [25]. Interpreting the biological meaning of the coordinates is fairly straightforward as well, as the magnitude of the gene loadings for each component identify the important genes, which can then be tested for enrichments in particular biological processes using tools like Enrichr [26] or StringDB [27].

While deep learning-based autoencoders are able to reconstruct gene expression with high fidelity, they fall short at steps (2) and (3) in the workflow described above. Both the relative importance of the different latent dimensions, and the biological meaning of these dimensions are opaque in deep autoencoders. "Interpretable" autoencoders (Fig. 1b) aim to improve step (3) for deep learning models by constraining the learned representation so that the latent dimensions correspond to known biological pathways. By "interpretable" autoencoder model, we refer to any model that learns a latent representation with dimensions that correspond to biological pathways or functions (Fig. 1b). These models can learn pathway embeddings that are complex, non-linear functions of the input genes, but are restricted in that each learned latent pathway dimension only incorporates information from genes that are pre-annotated to that pathway using databases like Reactome. This restriction is encoded either as a hard constraint using sparse masks across layers [22], or as a soft constraint using regularization [20] (see Methods Section 6.1 on "Model Architectures" for more details).

*While interpretable autoencoder models have latent nodes corresponding to biological pathways, there is no clear-cut way to identify which pathways are the most important in a dataset. Our approach, principled attribution for unsupervised gene expression analysis (PAUSE), aims to improve the utility of "interpretable" deep autoencoder models using techniques from the area of feature attribution (Fig. 1c). **While the eigenvalues in PCA correspond to the amount of variance in the original expression space explained by that component, deep learning-based autoencoders lack an obvious correspondence revealing how much variance is explained by each latent dimension.** Using approaches from game theory for credit allocation among players in cooperative games, we derive a novel pathway attribution that can be thought of analogously to the eigenvalues in PCA, in the sense that this attribution value shows how much variance in the original gene expression space is explained by each latent pathway. By posing the reduction in reconstruction error as the reward to be allocated in a cooperative game in which the pathways are the players (see Methods for more details), solution concepts like the Shapley value \cite{shapley1953value} or Aumann-Shapley value \cite{aumann2015values} can be used to provide **pathway attributions**. **While we primarily applied our pathway attributions to biologically-constrained models, they can be applied to any unsupervised autoencoder to identify the most important latent dimensions (see Supplementary Fig. 1).***

2. although the current manuscript is of the exactly same idea as the pmVAE work, the authors did not give a clear description and illustration as the pmVAE did. For example, figure 1 doesn't help the reader at all to understand the key idea of their methods and what is exactly the architecture of their autoencoder was also not provided in the method sections.

We thank the reviewer for suggesting that our concept figure could be modified to more clearly convey the primary focus of our paper. We completely redesigned the figure, which is included below. We believe Fig. 1b should more clearly describe sparse autoencoder architectures. More importantly, however, we believe Fig. 1a and 1c should more clearly sketch out the primary contribution of our paper by first sketching out the general workflow (Fig. 1a), then showing (Fig. 1c) how our novel unsupervised attribution works on step (2) in the workflow to rank the learned latent dimensions, and how gene attributions increase interpretability of the important dimensions.

a) Outline of unsupervised workflow

(1) Learn a **low-dimensional representation**

(2) **Rank the latent dimensions**

(3) **Interpret the important dimensions**

b) Interpretable autoencoders use sparse architectures to give latent dimensions biological interpretations

c) Principled attributions rank latent dimensions and aid in the interpretation of important dimensions

The new caption for this figure is also included for your convenience here:

Principled attributions complement biologically-structured networks to create more interpretable unsupervised models. **a**, An outline of a general workflow of unsupervised analysis of gene expression data, comparing classical linear approaches (left in each subpanel) and deep learning approaches (right in each subpanel). (1) First, a dimensionality reduction algorithm such as PCA (left) or a deep autoencoder (right) is applied to a dataset of gene expression values to learn a low-dimensional representation. (2) After this low-dimensional representation is learned, the learned dimensions must be ranked by their importance. This ranking is inherently provided in PCA, which sequentially maximizes directions of unexplained variance in the data. There currently are no principled approaches to provide this ranking in deep models, which is the gap in the literature filled by our novel loss attribution. (3) After finding the most important latent dimensions, the biological meaning of these dimensions is interpreted. In PCA (left), the contribution of different genes to each dimension can be found by examining the magnitude of the gene loadings. For deep learning models, feature attribution methods can be applied to determine gene contributions. **b**, In standard autoencoders, the learned latent variables have opaque meanings, as their relationships with input genes are unknown. Biologically-constrained models increase the interpretability of latent variables by using sparse connections or regularization to ensure that latent dimensions correspond to pre-defined pathways. **c**, We apply principled attribution methods to help rank the latent dimensions of autoencoder models and to interpret the biological meaning of the most important dimensions. Attributing the model's reconstruction error to the latent dimensions quantifies the importance of each latent dimension. Attributing the output of each latent dimension to the input genes quantifies the contribution of each input gene to each learned pathway.

Also, we thank the reviewer for their suggestion. We agree that the pmVAE architecture does play an important role in our paper, and that it should be better described in the main text. We have added several lines of text to try to add a more clear, plain language description of the idea behind the pmVAE architecture.

See manuscript lines 135-142:

The particular model architecture trained for this benchmark was a pathway module VAE (pmVAE), which is a sparse variational autoencoder model with deep, non-linear encoders and decoders \cite{gut2021pmvae}. **Unlike standard autoencoders, which have dense connections between every gene and each latent node in the first encoder layer, and dense connections between every latent node in each successive layer, the pmVAE uses sparse connections to define "modules" of nodes that have dense connections between nodes within the modules and no connections between nodes in different modules. The first layer of the encoder (and the last layer of the decoder) is a sparse layer that connects each gene only to the modules corresponding to the pathways to which it has been annotated. In this model, each module can be thought of as a separate dense autoencoder, which all sum together at the output layer.**

We have also added a new Supplementary Fig. 11 that includes additional text description of the cpmVAE architecture. The caption is attached here:

In order to enable our model to correct for unwanted sources of variation in gene expression data, it was necessary to modify the pmVAE architecture to be able to condition on nuisance variables. The architecture of our conditional pathway module VAE (cpmVAE) is similar to the pmVAE architecture, in that the internal nodes of the network are separated into modules corresponding to each pathway. Within each pathway module the intermediate nodes are all densely connected, and between each module there are no connections between nodes. In the bottleneck layer, while there is only one latent node per module depicted in the figure, there can be any number of bottleneck latent nodes per module. At the input and output layers, each module is connected to the input genes or the reconstruction genes with sparse connections ensuring that genes are only connected to the pathway modules that they are annotated to belong to. Dense modules are connected to all genes. The modification to allow for variable conditioning is to pass in a vector of additional covariates, which in addition to the genes are provided as an input to every module at both the input layer and again in the bottleneck layer.

3. The author only did very few tests on limited datasets. They should give more applications of their method in different scenarios to demonstrate its significance in the involved area.

We thank the reviewer for this suggestion to incorporate more datasets into our analyses. We have added three additional single cell RNA-seq datasets, which we analyze both by comparing the top pathways identified by unsupervised and supervised methods in these datasets, as well as running our “impute” and “retrain” benchmarks on these datasets.

We will first discuss our analysis of the dataset presented in McFarland et al. [ref. 68 in the manuscript], referred to here as “cancer cell lines”. In this dataset, the authors profile the gene expression of single cells from 24 cancer cell lines with either wild type or inactive mutant p53, which have all been treated with idasanutlin, a negative regulator of the tumor suppressor p53. Since all of the cells are stimulated with Nutlin, the primary signal is expected to be related to transcriptional regulation from p53. Our analysis of this dataset can be found below and in Supplementary Fig. 5.

When we first analyzed the top PAUSE attributions for a pmVAE model trained on this dataset (new Supplementary Fig. 5a), we observed that the top pathways we saw like Innate Immune System and Developmental Biology seemed to be more related to baseline differences between the tissues of origin of the cell lines, rather than differences in transcriptomic response and cell cycle arrest due to nutlin stimulation. When we visualize the entire latent space of the model using a t-SNE plot, we see that the latent space is in fact separated by different cell lines (Supplementary Fig. 5b). Specifically, if we look at one top pathway (“innate immune system”), we see that sample embeddings fall into distinct distributions based on cell line tissue type (Supplementary Fig. 5f). The “innate immune system” reactome pathway contains around 1,000 genes, and contains genes from signaling pathways like the PI3K/Akt pathway (a major pathway downstream of EGFR, known to be upregulated in certain cancers like lung cancer and colon cancer).

After observing this, we next use our modified *conditional* pathway module VAE (cpmVAE) to explicitly account for the presence of different cell lines in the data. By using the cpmVAE model here, we allow the embedding learned by our model to represent the biological variation of interest (here, nutlin stimulation and cell cycle arrest/apoptosis). As we can see in the new

Supplementary Fig. 5c, this use of the conditional model does succeed in removing separation by cell line from the latent space. Once cell line is accounted for in this way, we see that the PAUSE top pathways now represent sources of variation more relevant to the activity of TP53, such as the Reactome pathways “cell cycle” and “cell cycle mitotic”, both of which are noted in the original paper to be downregulated by idasanutlin treatment in the TP53 wild type cell lines. Notably, the pathways that are arguably most intuitively relevant to differences in TP53 status, “transcriptional regulation by TP53” and “TP53 regulates transcription of cell cycle genes”, are not top PAUSE pathways after conditioning on cell line. However, we note that these pathways do move up in the PAUSE attributions, from rank 34 to 27 and 77 to 33, respectively.

Inspection of top pathways found through the supervised analysis (new Supplementary Fig. 5e) shows both the advantage *and the drawback* of using labeled data to rank pathways. The supervised approach of LR score ranks the pathway “transcriptional regulation by TP53” as one of the most important pathways both after conditioning on the cell lines *and before conditioning on cell lines*. In both cases, this pathway is relevant to the stimulation of the cells by nutlin, but in the first case (see panel b), we can see that tissue-related differences are a larger source of variation than cell cycle signaling. This example also demonstrates the utility of PAUSE providing a *fully unsupervised* method that is capable of selecting pathways most relevant to transcriptomic variation without using labels. In this case, differences between cell lines represent an additional source of biological signal beyond the difference in TP53 status and nutlin stimulation. Using just the labels of the one specific biological signal to rank top pathways has the potential to miss other important sources of variation, which is why unsupervised attributions have an important use, even when labels do exist.

In addition to our analysis on the Cancer Cell Lines dataset, we also investigate a new dataset from Norman et al. [ref. 69 in the manuscript] to look at overlap between top pathways found with PAUSE attributions compared to a supervised approach, as done in Fig. 3. These new results can now be found in new Supplementary Fig. 6.

In this dataset, the authors use Perturb-seq (single-cell RNA-sequencing pooled CRISPR screens) to perturb single genes and pairs of genes in K562 cells and measure transcriptional response. In this way, the authors are able to quantitatively study genetic interactions (GIs) to map a manifold that describes the states a cell can occupy given perturbation. In order to observe a diverse range of GIs, the authors choose to perturb genes whose activation impacts growth of K562 cells, including cell cycle regulators, transcription factors, kinases, and phosphatases. The authors cluster the Perturb-seq profiles and annotate clusters including “erythroid”, “megakaryocyte”, and “granulocyte/apoptosis” based on overexpression of marker genes.

For our analyses, we apply a pmVAE model to a dataset consisting of unperturbed cells (serving as a control) as well as cells labeled as belonging to “granulocyte/apoptosis” gene perturbations due to their expression of granulocyte and apoptosis markers. We then use both PAUSE attributions and supervised (LR) attributions to analyze the top pathways, as done in Fig. 3.

We find that some pathways identified as important by both supervised and unsupervised techniques, like “innate immune system” and “neutrophil degranulation,” capture the expected variation between control cells and those perturbed to express granulocyte markers. In addition, other top PAUSE pathways that are not top supervised pathways, like “cell cycle” and “cell cycle mitotic” reflect the mechanism of the perturbation and the noted cell deaths caused by these perturbations. Many perturbed genes are cell cycle regulators, and the original authors show deviation of cell cycle as a clear effect of the perturbations. Supervised attributions also pick up on other pathways relevant to the “granulocyte” label, like “cytokine signaling,” but do not

include the transcriptomic variation due to cell cycle changes. Again, this example illustrates that supervised attributions are useful to the extent that we have labels defining the specific source of transcriptome variation we are interested in. To further explore other sources of variation, unsupervised attributions are of additional use.

Finally, as shown below and in the new Supplementary Fig. 2, we have added the results of our “impute” and “retrain” benchmark on the three additional single-cell RNA-seq datasets.

We run this benchmark on the cancer cell lines and K562 dataset, detailed above (Supplementary Fig. 5 and 6), as well as on a new dataset from Zheng et al. [ref. 70 in the manuscript]. This dataset, labeled here as “BMMC”, includes transcriptomic measurements of bone marrow mononuclear cells (BMMCs) from both healthy controls and patients with acute myeloid leukemia, taken before they underwent a bone marrow transplant.

In these additional benchmarks, we see the same patterns observed in Fig. 2 of the original manuscript: PAUSE attributions better identify the sources of transcriptomic variation in these datasets. In both the “impute” and “retrain” benchmark, PAUSE attributions significantly outperform other methods (two-sided Wilcoxon signed rank test, Bonferroni corrected $p=7.81 \times 10^{-3}$, statistic=0.0).

4. As a method or model work, it's unacceptable that no source codes are provided for others to use and to reproduce their results.

We thank the reviewer for pointing out that we had not provided a link to the codebase for PAUSE and the cpmVAE model. We indeed uploaded our codebase to our standard lab github, and included a link in section 6.11 of our biorXiv preprint in May

(<https://www.biorxiv.org/content/10.1101/2022.05.03.490535v1>), but forgot to add the link to our original manuscript. We deeply apologize for that.

We have fixed the problem now and updated the manuscript with links to source code available under an open source MIT license on Github (<https://github.com/suinleelab/PAUSE>), and have also uploaded the version of the code used in the paper to a Zenodo repository (doi:10.5281/zenodo.7331673). Accession information for the Github and Zenodo repositories are now listed in the Availability of Data and Materials section of the manuscript.

5. Additional minor comment: Figures 5a and 5b are not cited in order on page 13, where Figure 5b was cited first in the text.

Thank you, we revised the text of that section so that Fig. 5a is now mentioned first in the manuscript.

Reviewer #2: I recommend accepting this manuscript with minor revisions as suggested below.

In this manuscript, the authors have developed a pathway attribution method PAUSE that utilizes unsupervised analysis of gene expression data, along with pathway representations leading to more pathway-based biological interpretations than existing methods. In this manuscript, the authors have developed a pathway attribution method PAUSE that utilizes unsupervised analysis, along with pathway representations leading to more pathway-based biological interpretations of gene expression data. The strength of the method lies in pathway and gene level attributions that have been demonstrated to identify sources of transcriptomic variation and increase interpretability of the models respectively. The authors have demonstrated the utility of this method in identifying known and novel pathways showing most variation in 3 scRNA-Seq datasets with and without labeled data, as well identifying, and validating pathway and genes in a bulk RNA-Seq dataset.

Comments to author (in order of priority)

1. The paper emphasizes that PAUSE should complement supervised approaches for pathway identification. However, from Fig 3, the number of pathways unique to supervised approaches are limited. What is the justification for using these tools complementarily? It would be useful to have recommendations on how supervised approaches could be used in conjunction to PAUSE.

We thank the reviewer for this question. We believe that the justification for using these tools in a complementary fashion is that they ask fundamentally different questions of the data, and hence yield fundamentally different results. We have added an additional experiment with a new single cell RNA-seq dataset to demonstrate when and why these approaches find different top pathways.

We first train a pmVAE on the McFarland dataset [ref. 68 in the manuscript] a single cell RNA-seq dataset of a variety of cancer cell lines where the known ground truth perturbation measured is transcriptional response to nutlin stimulation in cell lines with either wild-type or mutant TP53. Nutlin should induce cell cycle arrest via MDM2 antagonism only in cell lines with wild-type TP53. When we look at top PAUSE attributions, we see that they do not identify cell cycle related pathways, and instead find pathways related to developmental origin like “developmental biology,” and pathways like “innate immune system,” which contains a large number of genes some related to signaling pathways like the PI3k-Akt pathway, a downstream signaling cascade from EGFR (important in colon/lung cancer). Plotting the full latent space of the model (using t-SNE), we can see that the major source of variation is related to differences in cell line tissue of origin, rather than nutlin stimulation. If you look at supervised attributions, they rank TP53 signaling regulation at the top, regardless of whether this is a top source of variation.

Our analysis of this dataset can be found below and in Supplementary Fig. 5.

This experiment highlights a situation where PAUSE attributions and supervised attributions rank pathways differently. There are arguments for either attribution representing the “right” solution here, depending on what you are trying to understand about the data. If you have a known stimulation and want to identify what expression differences are present between the two groups, supervised analysis returns you that answer. If you don’t have labels like this, and want to do exploratory analysis of a dataset to understand what sources of variation are present in this dataset

This experiment also allows us to demonstrate the power of our cpmVAE architecture. Imagine a scenario where the nutlin stimulation label is unknown, so supervised analysis would be impossible, but you did have access to the cell line information and knew that it was an unwanted source of variation to be controlled for. After conditioning on cell line information, when we generate PAUSE attributions the top pathways are related to cell cycle signaling. This shows that we can identify biologically relevant sources of variation *without any explicit need for variables*, if we just control for biologically irrelevant sources of variation and then run unsupervised analysis.

2. Please specify the link to the git repo for the method in the abstract.

We thank the reviewer for pointing out that we had not provided a link to the codebase for PAUSE and the cpmVAE model. We indeed uploaded our codebase to our standard lab github, and included a link in section 6.11 of our biorXiv preprint in May, 2022 (<https://www.biorxiv.org/content/10.1101/2022.05.03.490535v1>), but forgot to add the link to our original manuscript. We deeply apologize for that.

We have fixed the problem now and updated the manuscript with links to source code available under an open source MIT license on Github (<https://github.com/suinleelab/PAUSE>), and have also uploaded the version of the code used in the paper to a Zenodo repository ([doi:10.5281/zenodo.7331673](https://doi.org/10.5281/zenodo.7331673)). Accession information for the Github and Zenodo repositories are now listed in the Availability of Data and Materials section of the manuscript.

3. In section 2.5, the authors have accounted for batch effects from data source by modifying the pmVAE architecture. Based on Fig 5c, although a lot of the data source variation is corrected for, there still appear to be distinct clusters by batches. Please address how cpmVAE can address multiple known sources of variation without over-correcting for transcriptomic variation.

This is a great question, and we believe that it is difficult to formulate a general rule across all datasets and analyses. In our paper, we attempted to ensure that our cpmVAE model was correcting for unwanted variation without over-correcting for the real biological signal using the experiments in Supplementary Fig. 12. The goal of these experiments was to show that we were able to quantitatively diminish the latent space encoding of dataset source information (panel B), while still accurately reconstructing the observed expression (panel A), and retaining the information necessary in the latent space to predict ABeta IHC signal for the subset of panels for which ABeta labels were available (panel C, actually *improved* prediction). While these results

are certainly suggestive of an improvement in the control of nuisance variables and an improvement in *one* biological signal represented, they also do not necessarily demonstrate that *all* of the nuisance variation has been removed, or that *all* of the biological signal is optimally preserved (as there may be more signal related to biological processes not related to quantity of ABeta, which is not explicitly assessed here).

Some interesting related work on the problem of how much correction should be applied when trying to extract biological signatures from gene expression compendia is contained in a paper called “Correcting for experiment-specific variability in expression compendia can remove underlying signals.” The simulations in this paper show that choices surrounding correction depend upon the number of sources of variation being controlled, the size of the dataset in question, and the magnitude and structure of the signals to be extracted. Knowing information about the magnitude and structure of the signals of interest in real-world, non-synthetic datasets is almost always not possible, and would serve as interesting future work to help guide modeling decisions about nuisance factor correction.

We have added text to the section of the manuscript on “Sanity checks for bulk expression cpmVAE model” to address your point, see lines 654-663:

While these results are certainly suggestive of an improvement in the control of nuisance variables and an improvement in one biological signal represented, they also do not necessarily demonstrate that all of the nuisance variation has been removed, or that all of the biological signal is optimally preserved, as there are likely signals in the dataset from biological processes not related to density of A β , and the preservation of these signals is not explicitly assessed here. Prior work has been done on the problem of how much correction should be applied when trying to extract biological signatures from gene expression compendia \cite{lee2020correcting}. The simulations in this paper show that choices surrounding correction depend upon the number of sources of variation being controlled, the size of the dataset in question, and the magnitude and structure of the signals to be extracted. Knowing information about the magnitude and structure of the signals of interest in real-world, non-synthetic datasets is almost always not possible, and would serve as interesting future work to help guide modeling decisions about nuisance factor correction.

4. *Considering that the overlap between*

5. *Considering the limited overlap between supervised approaches and pathway attributions from PAUSE in sc datasets, how would those compare for AD bulk RNA-Seq data? Would you expect neuronal, neuroinflammatory pathways to be detected with supervised approaches?*

We thank the reviewer for this suggestion. While we initially did not run any supervised analysis on the AD bulk RNA-Seq data because of the limited number of labeled samples (664 samples, as compared to 1,288 in the Jurkat cell dataset or 13,576 in the PBMC dataset), we have now

added an experiment where we apply the “LR score” methodology to identify important pathways in the cpmVAE model trained on this data (see new Supplementary Table 1).

	lr_score
HALLMARK_MITOTIC_SPINDLE	0.049713
HALLMARK_NOTCH_SIGNALING	0.042565
HALLMARK_BILE_ACID_METABOLISM	0.040705
HALLMARK_PEROXISOME	0.039261
HALLMARK_TNFA_SIGNALING_VIA_NFKB	0.038146
HALLMARK_WNT_BETA_CATENIN_SIGNALING	0.038077
AUXILIARY	0.037853
HALLMARK_G2M_CHECKPOINT	0.036219
HALLMARK_P53_PATHWAY	0.036174
HALLMARK_REACTIVE_OXYGEN_SPECIES_PATHWAY	0.035841

Above are the top 10 pathways identified in the cpmVAE model by the LR score supervised method. We can see that the pathways identified by the LR score, in this case, do significantly overlap with the top pathways identified by our unsupervised attributions. For example, both supervised and unsupervised attributions identify pathways related to sources of oxidative stress: the “Reactive Oxygen Species” pathway and the “Peroxisome” pathway for the supervised attributions, and the “Mitochondrial Oxidative Phosphorylation” pathway for the unsupervised attributions. Not only are these pathways all related to potential sources of oxidative stress, but they also share overlapping genes. In particular, genes encoding different subunits of the NADH:ubiquinone oxidoreductase complex are present in both the “Mitochondrial Oxidative Phosphorylation” pathway and the “Reactive Oxygen Species” pathway.

The supervised and unsupervised approaches also overlap in identifying “TNF alpha signaling via NF-kB” as an important pathway in this dataset, which is, again, a key pathway known to mediate neuroinflammation.

We have added text to the manuscript to reflect this new experiment (see lines 664-675):

Finally, to compare the overlap between supervised approaches and pathway attributions from PAUSE on the AD dataset, we generated a ranked list of pathways using the supervised LR score method for the 664 brain expression samples with associated AB IHC protein density measurements (see Supplementary Table 1). We can see that the pathways identified by the LR score, in this case, do significantly overlap with the top pathways identified by our unsupervised attributions. For example, both supervised and unsupervised attributions identify pathways related to sources of oxidative stress: the Reactive Oxygen Species pathway and the Peroxisome pathway for the supervised attributions, and the Mitochondrial Oxidative Phosphorylation pathway for the unsupervised attributions. Not only are these pathways all related to potential sources of oxidative stress, but they

also share overlapping genes. In particular, genes encoding different subunits of the NADH:ubiquinone oxidoreductase complex are present in both the Mitochondrial Oxidative Phosphorylation pathway and the Reactive Oxygen Species pathway. The supervised and unsupervised approaches also overlap in identifying TNF alpha signaling via NF-kB as an important pathway in this dataset, which is, again, a key pathway known to mediate neuroinflammation.

6. *In section 2.4, you demonstrate that the densely connected unannotated nodes represent genes relevant to the ground truth for the perturbation of PBMCs with IFN-B. Please explain- 'we omitted particular pathways that should have been most relevant for the perturbation'. How were these pathways selected? Generally, for a perturbation, there is a cascade of downstream changes likely to occur. So, just removing IFN pathway may not be sufficient to show that the unannotated nodes represent ground truth genes- Supp Fig 3.*

Thank you for this question, we hope to clarify the experimental purpose of this Supplementary Figure. For the experiment where important pathways were omitted and then “re-discovered” using the unannotated nodes, it was not necessary to omit *all* ground truth pathways, or even the *most important* ground truth pathways. The point of this experiment was to show that densely connected unannotated nodes can represent pathways not present in the prior. We selected pathways that we knew would be relevant to the expression variation in the dataset by selecting all pathways with names containing the strings “IFN” or “interferon.”

We have added lines to the text that clarify that there is no guarantee that the dense modules represent the *most* important pathways, and that our experiment demonstrates more weakly that the dense modules can represent pathways missing from the prior that are truly important.

See lines 252-255:

While this experiment does not guarantee that the dense module will always find the most important sources of missing variation, it does demonstrate that it is able to represent important pathways when they are omitted from the prior, and that it will not necessarily represent unimportant pathways that are omitted from the prior.

7. *The computational usability of this method warrants more discussion- compute resources needed, compute time benchmarking etc.*

We thank the reviewer for bringing up the point of computational resources. We have recorded the compute time and resources needed for our analyses. We will focus our compute time analyses on a typical dataset used in these analyses, in this case, the Zheng et al. BMMC dataset [ref. 70 in the manuscript] where we look at 11,982 samples, each with measurements for 1,907 genes. In this case, we include 203 pathways in our sparse autoencoder. For model training, we

use one NVIDIA RTX A4000 GPU. The mean train time over 10 random initializations, with `max_epochs` set to 100, was 13.3 minutes. PAUSE attributions were calculated using the Path Explain repository (https://github.com/suinleelab/path_explain), which is a Python library for explaining feature importances and feature interactions in deep neural networks using path attribution methods [ref. 58–60 in the manuscript]. All attributions were generated using the “attributions” method of the PyTorch explainer; the argument “`num_samples`” was set to 200, while the argument “`use_expectation`” was set to “FALSE.” With this experimental setup, the mean time to compute PAUSE attributions was 22.54 seconds. In contrast, it took an average of 1.34 seconds to get supervised attributions using logistic regression.

We added this text to the methods section in manuscript lines 609-619.

8. The implementation of the PAUSE method also requires further discussion- 1) for the method to run effectively, what sample size of sc or bulk datasets are needed; 2) how do you recommend handling preprocessing and normalization of RNA-Seq data; 3) what method of imputation do you recommend for missing values; 4) Is this method fairly translatable to other omics datasets or integrative datasets etc.

1) For your first point, autoencoders for both single cell and bulk expression datasets have been shown to work successfully using a wide range of sample numbers. For example, Tan et al.’s 2016 paper “ADAGE-Based Integration of Publicly Available *Pseudomonas aeruginosa* Gene Expression Data with Denoising Autoencoders Illuminates Microbe-Host Interactions” successfully built denoising autoencoder models of *P. aeruginosa* expression using just 950 individual bulk expression microarray samples with measurements for 5,549 genes. Likewise, Chen et al.’s 2016 paper “Learning a hierarchical representation of the yeast transcriptomic machinery using an autoencoder model” also built an autoencoder model using 1,909 microarray samples. In 2018, Way & Greene trained a VAE on a pan-cancer RNA-seq dataset of >10K samples in their paper “Extracting a biologically relevant latent space from cancer transcriptomes with variational autoencoders.” Also in 2018, Lopez et al. applied their scVI VAE to single cell datasets of up to 1 million cells. Interestingly, they found there was an interplay between training length and samples, and for datasets with larger sample sizes, “fewer training iterations (or epochs) were needed.”

2 and 3) For your points 2 and 3, our paper does not necessarily take a position on the optimal approach for pre-processing and handling effects like dropout. In fact, we would argue that one of the strengths of our loss attribution is that it can be calculated for a variety of model architectures in a manner agnostic to their handling of dropout. For example, our manuscript predominantly worked with modifications of the pmVAE architecture, for which the authors pre-processed count data by log-normalization, then fit the data with an MSE loss. Other approaches like scVI and the deep count autoencoder believe it is preferable to leave the data as raw counts, then fit the data using negative binomial or zero-inflated negative binomial distribution, and claim that this type of modeling is a way to account for dropout.

To demonstrate that our pathway attributions can be calculated regardless of the loss used, we modified the pmVAE architecture to model count data (see the pmVAEModelNB class in

<https://github.com/suinleelab/PAUSE/blob/main/models.py>), then applied our loss attributions to that architecture (see https://github.com/suinleelab/PAUSE/blob/main/additional_notebooks/Negative%20Binomial%20Generative%20Model%20with%20Count%20Data.ipynb). We replaced the final linear layer of the pmVAE model with a Negative Binomial torch distribution layer, and added a gene-specific dispersion parameter to the model that is fit across all samples (similar to scVI, and adapted from their tutorial code at https://docs.scvi-tools.org/en/stable/tutorials/notebooks/module_user_guide.html). Then, rather than attribute the mean squared reconstruction error to the latent pathways, we attribute the average log likelihood of the data under the learned negative binomial distribution to the latent pathways.

See lines 498-503:

We also note that our method is agnostic to the exact form of the loss used to train the model. For example, approaches like scVI model RNA-seq data as discrete counts under a negative binomial (or zero-inflated negative binomial) distribution by optimizing the decoder to parameterize one of these distribution types. For a model like this, rather than using the mean squared reconstruction error as the function $\ell(x)$ in the PAUSE definition, one could instead use the negative log likelihood of the observed data under the model. A notebook demonstrating the application of PAUSE to an autoencoder parameterized in this manner is available in the “Additional Notebooks” section of the Github repository.

4) Your point number 4 is very interesting. While both the biologically-constrained architectures and loss attributions would be translatable to other datasets, the correct application in the context of a *multi-modal* omics dataset seems not entirely straight-forward. For example, take the case of ATAC-seq data. Unlike RNA-seq data, which reads off the sequence of expressed transcripts and consequently maps to individual genes, ATAC reads off the sequence of accessible chromatin, and may primarily map to intergenic regions and intronic regions of the genome (see, for example, Fig. 2d of <https://www.nature.com/articles/s41590-018-0273-1>). Even when ATAC maps to promoter sequences, these promoters may be bi-directional and control the transcription of multiple genes at once. Therefore, when choosing a biologically-constrained representation for ATAC-seq data, it may be more appropriate to cluster the bin features on the basis of the TF-binding motifs contained within those bins.

If you wanted to build a multi-modal autoencoder with a shared latent space for both RNA- and ATAC-seq data, rather than choosing REACTOME pathways, it might be better to group features on the basis of TFs, using information like the information present in the ChEA enrichment tool, which defines TF-linked gene sets on the basis of “putative targets as determined by CHIP-seq experiments from ENCODE, ReMap, and individual publications; co-expression of TFs with other genes based on processed RNA-seq from GTEx and ARCHS4; co-occurrence of TFs with

other genes by examining thousands of gene lists submitted to the tool Enrichr; and gene signatures resulting from single TF perturbations followed by genome-wide gene expression experiments.” This could allow the RNA-seq features and ATAC-seq features to share a single latent space.

This would then leave the interesting problem of figuring out how to calculate attributions. Should the reconstruction errors for each individual feature get equal weight? Or should multiple sets of “TF/Pathway” attributions be calculated and compared? For example, one could imagine calculating loss attributions using the reconstruction error for RNA-seq, then comparing these to the reconstruction error for ATAC-seq. This might reveal biologically interesting patterns, such as TFs associated with BOTH clear gene expression patterns AND clear changes in chromatin accessibility. On the other hand, there might be other TFs that explain a large amount of expression variation but not chromatin accessibility and vice versa. The biological significance of these discordant TFs might be interesting.

In conclusion, we believe that this would be the perfect topic for follow-up work, and would be technically easy to implement given the code written for this paper, but biologically interesting to analyze. We have therefore added additional sentences to the discussion section to indicate that this would be interesting and possible future work.

See lines 373-378:

Even in biology, pathway annotations may not represent the most useful representation. In particular, translating interpretable approaches to multi-modal datasets will be an important future direction. Defining interpretable representations capable of grouping features across data modalities will be important to integrate these diverse datasets. For example, one could design a transcription factor-based latent space, linking both the expression measurements of genes annotated to be regulated by each transcription factor, and the chromatin accessibility of genome regions containing sequence motifs related to the binding of each transcription factor.

9. *In Fig 4, how are the 'most important' genes from each node defined- please mention this briefly in the figure legend.*

Thank you for this suggestion, we have updated the figure legend to describe how the most important genes from each node are defined.

Fig 4 legend, new additions are in bold:

b-c, Gene attribution dependence plots for the two most important genes, **ranked by average magnitude gene attribution over all samples in the dataset**, for TCR Signaling Latent Node 1.

d-e, Gene attribution dependence plots for the two most important genes, **ranked by average magnitude gene attribution over all samples in the dataset**, for TCR Signaling Latent Node 2.

Reviewer #3: In this paper, the authors develop a method PAUSE that enables one to interpret unsupervised deep learning models, primarily autoencoders, used to model gene expression data from bulk and single cell assays. PAUSE builds on top of an existing framework called pathway module VAE (pmVAE) and is extended to incorporate also batch effects and batch removal in a conditional pmVAE (cpmVAE). The pmVAE model incorporates known gene-pathway memberships while learning the underlying latent space. PAUSE uses a new feature attribution method that identifies pathways that are most useful for explaining the overall variation in the data, as well as individual genes in each pathway that are the major contributors to the overall expression variation. PAUSE is applied to both single cell datasets with known class labels as well as bulk dataset with known batch effects. PAUSE was able to find the major sources of variation and is able to better rank potential pathways (as assessed by the impute and retrain metrics) from single cell data. On bulk data for various AD patient samples PAUSE identifying a mitochondria related pathway which was validated using a C. elegans model. The paper is well written and does a very nice job of contextualizing their work in light of the current literature. The paper also addresses an increasingly important problem of interpretability of ML models. I have a few questions/comments about the advantage of this approach over the pmVAE approach and simpler approaches and some claims in the paper that need additional clarification with some experiments.

1. Although the paper is well written it is initially a bit misleading as to what the proposed method is doing. Initially I thought the authors are developing a new unsupervised learning method for modeling gene expression data which has the automatic ability to also find informative pathways and that might not need the information of existing pathways. However, what the contribution of this paper is really a new feature attribution method which builds on top of an existing framework. This part of the paper needs to be better described. I feel the "Overview of PAUSE" section right before the results should talk in more detail about the pmVAE and cpmVAE models. Even the methods don't really go into much details about the pmVAE model. To understand what PAUSE is doing, we need to understand pmVAE and its various model components. It would be helpful to describe this better.

We have added more text describing the pmVAE model in our results section.

See lines 135-142:

The particular model architecture trained for this benchmark was a pathway module VAE (pmVAE), which is a sparse variational autoencoder model with deep, non-linear encoders and decoders \cite{gut2021pmvae}. Unlike standard autoencoders, which have dense connections between every gene and each latent node in the first encoder layer, and dense connections between every latent node in each successive layer, the pmVAE uses sparse connections to define “modules” of nodes that have dense connections between nodes within the modules and no connections between nodes in different modules. The first layer of the encoder (and the last layer of the decoder) is a sparse layer that connects each genes only to the modules corresponding to the pathways to which it has been annotated. In this model, each module can be thought of as a separate dense autoencoder, which all sum together at the output layer.

We have also completely re-designed our concept figure. This concept figure has a new panel that more clearly describes the biologically-constrained model architecture. This concept figure also has been redesigned to do a better job of highlighting the primary contribution of our paper, which you correctly describe above. We hope it is no longer initially misleading or confusing. (Revised concept figure, Fig. 1, attached below).

a) Outline of unsupervised workflow

(1) Learn a **low-dimensional representation**

(2) **Rank the latent dimensions**

(3) **Interpret the important dimensions**

b) Interpretable autoencoders use sparse architectures to give latent dimensions biological interpretations

c) Principled attributions rank latent dimensions and aid in the interpretation of important dimensions

The new caption for this figure is also included for your convenience here:

Principled attributions complement biologically-structured networks to create more interpretable unsupervised models. **a**, An outline of a general workflow of unsupervised analysis of gene expression data, comparing classical linear approaches (left in each subpanel) and deep learning approaches (right in each subpanel). (1) First, a dimensionality reduction algorithm such as PCA (left) or a deep autoencoder (right) is applied to a dataset of gene expression values to learn a low-dimensional representation. (2) After this low-dimensional representation is learned, the learned dimensions must be ranked by their importance. This ranking is inherently provided in PCA, which sequentially maximizes directions of unexplained variance in the data. There currently are no principled approaches to provide this ranking in deep models, which is the gap in the literature filled by our novel loss attribution. (3) After finding the most important latent dimensions, the biological meaning of these dimensions is interpreted. In PCA (left), the contribution of different genes to each dimension can be found by examining the magnitude of the gene loadings. For deep learning models, feature attribution methods can be applied to determine gene contributions. **b**, In standard autoencoders, the learned latent variables have opaque meanings, as their relationships with input genes are unknown. Biologically-constrained models increase the interpretability of latent variables by using sparse connections or regularization to ensure that latent dimensions correspond to pre-defined pathways. **c**, We apply principled attribution methods to help rank the latent dimensions of autoencoder models and to interpret the biological meaning of the most important dimensions. Attributing the model's reconstruction error to the latent dimensions quantifies the importance of each latent dimension. Attributing the output of each latent dimension to the input genes quantifies the contribution of each input gene to each learned pathway.

Finally, we have totally re-written the first two sections of the paper to more clearly describe the major contributions of the paper and give all of the necessary context to understand these contributions.

2. The comparison of the specific pathways that are identified using the PAUSE versus supervised methods could be more compelling. While the authors discuss the pathways not identified by PAUSE, what would be even more compelling is if the authors could discuss pathways that only PAUSE identifies and that these are indeed relevant to what the supervised methods are giving us.

We thank the reviewer for this question. We believe that the justification for using these tools in a complementary fashion is that they ask fundamentally different questions of the data, and hence yield fundamentally different results. We have added an additional experiment with a new single cell RNA-seq dataset to demonstrate when and why these approaches find different top pathways.

We first train a pmVAE on the McFarland dataset [ref. 68 in the manuscript], a single cell RNA-seq dataset of a variety of cancer cell lines where the known ground truth perturbation measured is transcriptional response to nultin stimulation in cell lines with either wild-type or mutant TP53. Nutlin should induce cell cycle arrest via MDM2 antagonism only in cell lines with wild-type TP53. When we look at top PAUSE attributions, we see that they do not identify cell

cycle related pathways, and instead find pathways related to developmental origin like “developmental biology,” and pathways like “innate immune system,” which contains a large number of genes some related to signaling pathways like the PI3k-Akt pathway, a downstream signaling cascade from EGFR (important in colon/lung cancer). Plotting the full latent space of the model (using t-SNE), we can see that the major source of variation *IS* related to differences in cell line tissue of origin, rather than nutlin stimulation. If you look at supervised attributions, they rank TP53 signaling regulation at the top, regardless of whether this is a top source of variation.

Our analysis of this dataset can be found below and in Supplementary Fig. 5.

This experiment highlights a situation where PAUSE attributions and supervised attributions rank pathways differently. There are arguments for either attribution representing the “right” solution here, depending on what you are trying to understand about the data. If you have a known stimulation and want to identify what expression differences are present between the two groups, supervised analysis returns you that answer. If you don’t have labels like this, and want

to do exploratory analysis of a dataset to understand *what sources of variation are present in this dataset*, PAUSE provides a more accurate view.

This experiment also allows us to demonstrate the power of our cpmVAE architecture. Imagine a scenario where the nutlin stimulation label is unknown, so supervised analysis would be impossible, but you did have access to the cell line information and knew that it was an unwanted source of variation to be controlled for. After conditioning on cell line information, when we generate PAUSE attributions the top pathways are related to cell cycle signaling. This shows that we can identify biologically relevant sources of variation *without any explicit need for variables*, if we just control for biologically irrelevant sources of variation and then run unsupervised analysis (see Supplementary Fig. 5 and Main Fig. 5).

3. The gene attribution aspect of PAUSE helps to further add interpretability, but I feel this section could be better described and have more results. First, it would be helpful to more clearly describe perhaps upfront what is a node in a pathway. Is this a latent factor or is it a gene in the pathway. The model is essentially based on pmVAE so it would be very helpful to describe the pmVAE model in more detail and what nodes really correspond to in the model. It is also not clear what the authors meant by "removing a pathway". Does it mean it was removed from the input/architecture of the VAE? Where exactly is a dense node added in the model?

Thank you for these suggestions, we have added several clarifications to the text. First, we add more text clarifying that when we discuss pathway nodes in the paper, we are referring to the nodes in the bottleneck layer of the corresponding pathway module unless otherwise specified. We then clarify that in this experiment when we “remove” a pathway, this pathway is omitted by training a *new model* without the corresponding modules for those pathways.

See manuscript lines 234-243:

*In addition to differentiating sub-processes within an annotated pathway module, another use for gene attributions is identifying biological processes learned by unannotated, densely-connected **modules**. In addition to sparse modules corresponding to annotated biological pathways, densely connected modules can be jointly modeled to capture novel biology. In order to verify that densely connected **modules** could identify biological expression programs not represented in prior knowledge bases, we modeled a dataset where we know the ground truth perturbations (PBMCs stimulated with IFN- β). **We omitted a group of pathways related to the known perturbation by training a new model without the corresponding modules for those pathways**, but included a module **connected to all genes with four nodes in its bottleneck layer**. When we used gene attributions to identify the important expression contributors for the **latent nodes in the bottleneck layer of this densely-connected, unannotated module**, we found that they corresponded to the expected ground truth biology (Supplementary Fig. \ref{fig:if_top_genes}).*

We have also added a new experiment that we believe demonstrates the importance and utility of gene attributions. In this revision, we have added several experiments to show that our unsupervised pathway attributions are useful even when used with unannotated, standard VAEs (see response to your Point 5 below). After our unsupervised loss attributions identify the top pathways, since these pathways are not biologically-constrained, gene attributions become even *more* important for the biological interpretation of these pathways.

4. Related to 3., the finding of the correlated and uncorrelated gene pairs is interesting, but this is shown only for one pair of genes. Fig 4D is a very common scatter plot for two genes that are somewhat correlated. I am not sure how we can interpret the specific cell populations that exhibit correlation vs not. Is it possible to identify these populations from the latent factors? Overall, the results here felt a bit anecdotal and more comprehensive analysis of gene attribution could be done.

Thank you, in response to this suggestion we have conducted a more comprehensive analysis of the gene attributions in Fig. 4. We believe this new analysis shows the importance of including gene attributions when multiple nodes are included in the bottleneck layer of each pathway module. In the original pmVAE paper, the authors noted that using 4 nodes in the bottleneck layer improved separation of the Jurkat cells according to CD3/CD28 stimulation status when compared to 1 node in the bottleneck layer of each pathway module. However, the authors provide no way to identify the biological significance of the different nodes learned within the same pathway. Updated Fig. 4 and manuscript section (new figure panel is 4f) attached here for your convenience:

We updated the text in the manuscript to describe this experiment (see lines 222-232, changes in bold):

We see that Node 1's output is high when PTPRC's expression is high (Fig. 4b), and low when PTPN22 expression is high (Fig. 4c). This means that in cells where PTPRC and PTPN22 are co-expressed, the contributions of these two genes will cancel out and the latent node's activation will have a low magnitude. Node 2's output, in contrast, is highly negative when PTPRC expression is high (Fig. 4e) and highly negative when PTPN22 expression is high (Fig. 4f). **When we visualize all of the cells in the dataset by their embedding values for these two latent nodes after filtering out cells where the values of these top genes are dropped out (see Fig. 4f), we see that Node 2 separates anti-CD3 and anti-CD28 antibody-stimulated cells (Wilcoxon rank-sums test statistic $= -6.27$, pvalue $= 3.39 \times 10^{-10}$), while Node 1 does not (Wilcoxon rank-sums test statistic $= -0.95$, pvalue $= 0.342$).** This demonstrates the utility of gene attributions. While previous work found that having multiple nodes in the bottleneck layer of each pathway module led to embeddings that were more discriminative for T cell activation [cite{gut2021pmvae}], there was no direct way to distinguish which expression patterns the model was using to define each latent node.

Our new figure better demonstrates how gene attributions can help clarify the biological significance of the learned representation. We know that the model has learned that there are

multiple axes of variation related to the expression of genes within this pathway from Fig. 4a, as the values of these latent dimensions are not highly correlated across the cells in the dataset. The gene attributions previously generated in Fig. 4 (see panels b-e above) demonstrate the very different expression patterns learned by the two latent variables. The first latent node has a high activation when PTPRC/PTPN22 are *uncorrelated*, allowing the model to represent cells where these two genes are regulated independently. The second latent node has a high activation when PTPRC/PTPN22 are *highly correlated*, allowing the model to represent cells where these genes are highly co-regulated.

Without a deeper analysis, just from looking at these attribution plots, one might hypothesize that Node 2 may represent general, non-specific changes in expression of this pathway, as genes are co-regulated and have diffusely greater or lesser expression. One might hypothesize that Node 1 could represent the specific up or down- regulation of genes that specifically positively or negatively regulate T cell activation. PTPN22 is a negative regulator of T cell activation (<https://www.ncbi.nlm.nih.gov/pmc/articles/PMC6402334/>), while PTPRC tends to have more positive regulatory roles in T cell activation (<https://www.ncbi.nlm.nih.gov/pmc/articles/PMC8380896/>), so it is reasonable to expect that an expression pattern that allows for simultaneously high PTPRC and low PTPN22 might be more closely aligned with T cell activation, as the effects of these genes may be less likely to cancel out.

Rather than looking at the scatterplot of two genes in Fig. 4f that we had initially shown in this figure, in our new revised experiment we demonstrate here that we can get a better understanding of our model by combining supervised information with our gene attribution information. When we visualize the first two learned latent dimensions of the TCR Signaling module and color-code each cell by whether or not it had been stimulated by anti-CD3/anti-CD28 antibodies (after filtering out cells where PTPN22 and PTPRC expression were dropped out), we can see that it is actually **Node 2** that separates the activated cells and not Node 1. This indicates that after being stimulated by anti-CD3/anti-CD28 antibodies, the expression of genes in the TCR signaling pathway broadly go up, and it is not just positive regulators of T cell activation whose expression is increased.

5. The authors say that the approach is broadly applicable to any unsupervised (auto-encoder)-based method. However, PAUSE is demonstrate primarily with pmVAE and it is not clear how well PAUSE would work for VAEs with less structure or prior knowledge. Some more experiments could demonstrate the generality of PAUSE. For example, one could use more noisy or incomplete pathway information and see how much PAUSE's prioritization schemes are affected. It would be nice to show PAUSE with other types of VAEs as well.

While we primarily generated PAUSE attributions for pmVAE models in our original manuscript, we did also generate pathway attributions for a VAE with a linear decoder (in the style of LDVAE <https://academic.oup.com/bioinformatics/article/36/11/3418/5807606>, and Interpretable AE <https://www.biorxiv.org/content/10.1101/2020.12.02.401182v1>), in what is now Supplementary Fig. 3.

We agree, however, that to claim our approach is “broadly applicable,” we should apply it to even more additional autoencoder types. Therefore, for this revision, we first added an experiment where we apply PAUSE pathway attributions to a VAE with *no* prior knowledge encoding structure. We compare this side-by-side to a standard workflow done with PCA (see Datlinger et al. “Pooled CRISPR screening with single-cell transcriptome readout” (2017) and Macosko et al. “Highly parallel genome-wide expression profiling of individual cells using nanoliter droplets” (2015)), demonstrating how this classical interpretable workflow is enabled for deep learning models.

We first apply both PCA (scikit-learn implementation, with `n_components = 10`), and a VAE (10 latent nodes in the bottleneck layer), to the Jurkat cell T cell activation dataset. After training both of these models, we want to know which of the latent nodes in the bottleneck layer or principal components were most important. We can use the “`explained_variance_`” method from the scikit-learn implementation of PCA to calculate the contribution of each PC to the reconstruction. Our PAUSE pathway attributions can then provide the analogous calculation for the VAE model. Comparing these “attributions” between PCA and the VAE reveals interesting differences between the models learned. For example, the VAE learns a sparser model than PCA, representing a greater proportion of the observed gene expression variation in a smaller number of latent variables (see panels a and b in new Supplementary Fig. 1, included below, fraction of total variance explained by each principal component from PCA on top, fraction of total variance explained by latent variables from VAE on bottom).

We can then create a scatter plot of the learned embedding for the two most important PCs or LVs, and use gene attributions to interpret the biological significance of these dimensions (see panels c/d). We generate the top genes for each PC (using the magnitude of the gene loadings) and the top genes for each VAE LV using our integrated gradients-based gene attribution method (average magnitude IG attributions for the learned mean of the LV averaged over all cells in the dataset). We then take the top 100 genes and check for enrichments in biological processes using the enrichment testing in the web tool StringDB (see Supplementary Table 2 for exhaustive list of pathway enrichments found). We find that the top PC is mostly enriched for cell cycle-related genes, while the second PC is enriched for both cell-cycle related genes but also T cell activation pathway genes. Interestingly, for both of the top VAE LVs, there are many enrichments for both cell cycle processes AND T cell activation processes.

These results highlight a difference between biologically-constrained models of gene expression and unconstrained models. The PCs or latent variables in unconstrained models represent the expression of multiple biological processes simultaneously, possibly indicating a common regulation of these processes, or a hierarchical relationship between them (e.g. downstream effects of one process or pathway on another). These results also highlight the utility of pathway and gene attributions for unsupervised analysis of gene expression. The type of workflow done with PCA in the past (learn PCs/LVs, find the important ones, identify which genes contribute to these PCs) was not possible with deep neural autoencoders, and is enabled by our PAUSE method.

See new Supplementary Fig. 1, included below:

Caption for Supplementary Fig. 1:

PAUSE Loss Attribution enables analogous workflow to classical models for standard VAEs. **a**, The fraction of model variance explained by each of the top ten principal components of the Jurkat cell scRNA-seq dataset \cite{datlinger_pooled_2017}, calculated using the `'explained_variance_'` method in the scikit-learn implementation of PCA. Each fraction is equal to the corresponding eigenvalue of the covariance matrix of the original data, divided by the sum of all of the eigenvalues. **b**, The fraction of model variance explained by each of the ten latent dimensions learned by a variational autoencoder trained on the Jurkat cell scRNA-seq dataset, calculated using the global PAUSE loss attribution (averaged across all samples in the dataset), divided by the sum of all of the global loss attributions. **c**, A visualization of the Jurkat cell dataset embedded by the top two

*principal components. The biological annotations for each of the axes are determined by examining the biological processes enriched in the top 100 magnitude genes in gene loadings for these components, calculated using the enrichments tool from the StringDB web server. **d**, A visualization of the Jurkat cell dataset embedded by the two most important latent dimensions learned by the VAE. The biological annotations for each of the axes are determined by examining the biological processes enriched in the top 100 average magnitude gene attributions for these dimensions, calculated using the enrichments tool on the StringDB web server. A list of all enriched pathways for the principal components and latent variables in **c** and **d** can be found in Supplementary Table 2.*

To ensure the validity of our unsupervised loss attributions on unconstrained autoencoders like the VAE used in the above experiment, we re-ran our benchmark experiment using the datasets from Fig. 2, but instead of using a pmVAE model, we use an unconstrained VAE (see below and in new Supplementary Fig. 4). These results ensure that our PAUSE attributions are actually identifying the latent variables responsible for the variation in the model. Interestingly, we observe that on standard VAEs without any latent space constraints (only optimized to minimize divergence from an isotropic Gaussian prior and reconstruct input), and optimized with the weight of the penalty for the KL divergence equal to the reconstruction error (in contrast to pmVAE, which decreased the penalty weight for KL divergence), we see that KL divergence does a significantly better job of identifying important pathways. PAUSE still significantly outperforms KL Divergence, but on this one type of autoencoder model (standard VAE), KL is a reasonably performant method.

Finally, in addition to demonstrating the validity and utility of our unsupervised attributions on VAEs without biological constraints, we also wanted to demonstrate the broad applicability of our loss attributions to models trained with different types of losses. Our manuscript predominantly worked with modifications of the pmVAE architecture, for which the authors pre-processed scRNA-seq count data by log-normalization, then fit the data with an MSE loss. Other approaches like scVI and the deep count autoencoder believe it is preferable to leave the data as raw counts, then fit the data using negative binomial or zero-inflated negative binomial distribution, and claim that this type of modeling is a way to account for dropout.

To demonstrate that our pathway attributions can be calculated regardless of the loss used, we modified the pmVAE architecture to model count data (see the pmVAEModelNB class in <https://github.com/suinleelab/PAUSE/blob/main/models.py>), then applied our loss

attributions to that architecture (see https://github.com/suinleelab/PAUSE/blob/main/additional_notebooks/Negative%20Binomial%20Generative%20Model%20with%20Count%20Data.ipynb). We replaced the final linear layer of the pmVAE model with a Negative Binomial torch distribution layer, and added a gene-specific dispersion parameter to the model that is fit across all samples (similar to scVI, and adapted from their tutorial code at https://docs.scvi-tools.org/en/stable/tutorials/notebooks/module_user_guide.html). Then, rather than attribute the mean squared reconstruction error to the latent pathways, we attribute the average log likelihood of the data under the learned negative binomial distribution to the latent pathways.

See Methods section, lines 498-503:

We also note that our method is agnostic to the exact form of the loss used to train the model. For example, approaches like scVI model RNA-seq data as discrete counts under a negative binomial (or zero-inflated negative binomial) distribution by optimizing the decoder to parameterize one of these distribution types. For a model like this, rather than using the mean squared reconstruction error as the function $\ell(x)$ in the PAUSE definition, one could instead use the negative log likelihood of the observed data under the model. A notebook demonstrating the application of PAUSE to an autoencoder parameterized in this manner is available in the “Additional Notebooks” section of the Github repository.

6. The authors applied their method to an AD expression dataset and identified Oxidative phosphorylation and several mitochondrial genes that might be involved. A C elegans model of AD was used to show that these genes have a phenotype when knocked down. These experiments could be described in detail. It was not clear based on the ranking of the genes, why one would see the increase in time for paralysis onset. What does this mean in terms of the role of the mitochondria in AD? Are these genes overexpressed in some way in AD patients?

We thank the review for this insightful question. Our unsupervised attributions do not predict the directionality of effect from the human datasets, so we instead relied on the *in vivo* animal model to address this question. Computationally inferring directionality also would not necessarily have been a successful strategy, as the changes in observational data of gene expression in the brain could reflect either cause or effect. For example, increased expression of a particular gene in an AD brain could occur because that gene causes (or contributes to AD) or it could occur because that gene is induced in response to AD-associated pathology (perhaps as a protective effect). In this hypothetical case, if the former model is correct, then knocking the gene down may be protective, but in the latter model knocking the gene down may be detrimental and accelerate disease.

In the specific case of the oxidative phosphorylation electron transport chain (ETC) components, we observed that when we made the intervention of RNAi knockdown, it conferred protection against amyloid beta toxicity, as evidenced by delayed paralysis in the experiments in Fig. 6.

This is most immediately supportive of the hypothesis that normal ETC activity contributes to the pathological consequences (in this case, paralysis) of high amyloid beta burden. Alternatively, however, knock-down of ETC components could induce a protective, compensatory response that attenuates amyloid beta toxicity. Because we do not measure all of the expression changes in the worms after knocking down ETC components (only validating that ETC is actually knocked down), we cannot guarantee that ETC is the *only* process mediating ABeta proteotoxicity, or even guarantee its direction. While our experiment does not fully elucidate the complete mechanism of ABeta toxicity, our novel findings do demonstrate that ***ETC components play an important role in mediating this process***. These results provide a starting point for future pharmacological research and further elucidation of the complete mechanism, which will be an important topic to address experimentally through future study.

To clarify this in the manuscript, we have made the following changes:

See revised manuscript lines 310-318 (Manuscript changes are in bold):

*To gain insight into the biological relevance of the genes identified by our PAUSE analysis of the Alzheimer's brain expression data, we used the nematode *C. elegans*, a well-established animal model of A β proteotoxicity. We conducted experiments with GMC101, a transgenic worm line displaying an age-associated aggregation of human A β 1-42 peptide resulting in rapid onset of age-associated paralysis. A stringent reciprocal best hits (RBH) approach (BLAST *e*-value < 10⁻³⁰; details in Methods) was used to identify nematode orthologs for human genes. **This assay can be used to test the effects of various genes on A β proteotoxicity, as transgenic A β 1-42 accumulates over time in the body wall muscle of GMC101 worms, leading to a paralysis phenotype mediated by A β proteotoxicity. Gene expression perturbations that significantly increase the observed time to paralysis therefore are demonstrated to impact A β 1-42 toxicity in this model organism.***

See revised manuscript lines 324-327:

We observed that RNAi knockdown conferred protection against amyloid beta toxicity, as evidenced by delayed paralysis. This may indicate that mitochondrial Complex I activity contributes to the pathological consequences of high A β burden. Alternatively, knock-down of mitochondrial Complex I components could induce a protective response that attenuates A β toxicity. This will be an important topic to address experimentally through future study.

7. I could not find a dedicated codebase for PAUSE and the cpmVAE model. The linked page https://github.com/suinleelab/path_explain seems to be associated with an already published paper. This needs to be clarified.

We thank the reviewer for pointing out that we had not provided a link to the codebase for PAUSE and the cpmVAE model. We indeed uploaded our codebase to our standard lab github, and included a link in section 6.11 of our biorXiv preprint in May of this year (<https://www.biorxiv.org/content/10.1101/2022.05.03.490535v1>), but forgot to add the link to our original manuscript. We deeply apologize for that.

We have fixed the problem now and updated the manuscript with links to source code available under an open source MIT license on Github (<https://github.com/suinleelab/PAUSE>), and have also uploaded the version of the code used in the paper to a Zenodo repository ([doi:10.5281/zenodo.7331673](https://doi.org/10.5281/zenodo.7331673)). Accession information for the Github and Zenodo repositories are now listed in the Availability of Data and Materials section of the manuscript.

To clarify the purpose of the software that was linked in the previous version of the manuscript, the link in the last version of the manuscript is to the “Path Explain” repository, a general purpose piece of software that we wrote for a previous paper that can be used to calculate the path integrals necessary for *many* different feature attribution methods. We originally wrote this software to calculate a feature interaction value known as Integrated Hessians, but this software can be used to calculate other attributions, like Integrated Gradients, Expected Gradients, Expected Hessians, Input Gradients, and now our newly proposed *unsupervised* pathway attribution method PAUSE.

Second round of review

Reviewer 1

In this revision, the authors have made great efforts to improve the manuscript according to the reviewers' concerns. Especially, they add a new figure (Figure 1) with detailed introductions to more clearly clarify the significant contributions of this work compared to other existing works. Moreover, they also used more datasets and case studies to demonstrate the applicability of their method. Overall, they addressed all my major concerns.

However, although the author updated the link to the codes, the GitHub repo is organized in a way far to meet the expectation of a method GitHub repo. The authors should further format the GitHub repo to make it friendlier for readers to use their method and reproduce their results.

Specifically,

1) for the model, the author should provide detailed documentation on how to use their code with example datasets. As a wide-used routine for method repositories, a step-by-step manual with parameters (or arguments) introductions listed should be given on the README page of the GitHub repo.

2) for the reproducibility sources, the author should provide the codes with the direct data. Currently, the author only provides the codes without the data, it's hard for the readers to reproduce their results, like figures or tables. If the data exceed the size limit of GitHub, alternative repo providers, such as the figShare or Zenodo, should be used. Currently, the author just made a copy of their GitHub repo to Zenodo. It is not necessary to just do that without providing the data for reproducibility.

Reviewer 3

The authors have sufficiently addressed my comments. I don't have further questions.